# NaviAgent: Graph-Driven Bilevel Planning for Scalable Tool Orchestration

Yan Jiang [* 1]  Hao Zhou [* 1]  Lizhong GU [1]  Tianlong Li [1]  Ruinan Jin [2]  Wanqi Zhou [1]  Ai Han [✉ 1]

## Abstract

Large Language Models (LLMs) increasingly act as function-call agents that invoke external tools to tackle tasks beyond their static knowledge. However, they typically invoke tools one at a time without a global view of task structure. As tools often depend on one another, this leads to error accumulation and poor scalability, particularly when scaling to hundreds or thousands of tools. To address these limitations, we propose NaviAgent, an explicit bilevel architecture that decouples task planning from tool execution through graph-based modeling of tool relations. At the planning level, the LLM-based agent decides whether to respond directly, clarify intent, or retrieve and execute a toolchain independent of inter-tool complexity. At the execution level, a Tool World Navigation Model (TWNM) encodes structural and behavioral relations among tools, steering the agent to compose scalable and robust invocation sequences. Incorporating feedback from real tool interactions, NaviAgent achieves closed-loop alignment between planning and execution, enabling adaptive navigation in large-scale tool ecosystems. Evaluations on API-Bank and ToolBench show consistent improvements in task success rate (TSR), with TWNM yielding an average gain of 13.1 points on complex tasks. Further tests on 50 real APIs across 7 domains show consistent gains of 4.3–12.0 points, with fewer steps and latency, demonstrating robust generalization under real-world dynamics.

## 1. Introduction

Large language models (LLMs) are growing in adoption as function call agents, shifting from simple tool invocation

[*]Equal contribution [1]JD.COM, China [2]Department of Electrical and Computer Engineering, The Ohio State University, USA. Correspondence to: Ai Han <hanai5@jd.com>.

*Proceedings of the 43$^{rd}$ International Conference on Machine Learning*, Seoul, South Korea. PMLR 306, 2026. Copyright 2026 by the author(s).

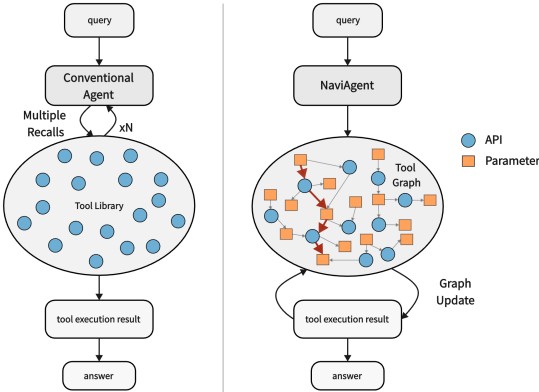

*Figure 1.* Conventional function call agents vs. NaviAgent.

to supporting intricate multi-stage workflows (Shen et al., 2023; Yang et al., 2023; Qu et al., 2025). In real-world environments, however, agents must operate over large, heterogeneous, and continually evolving tool ecosystems. The resulting combinatorial growth of possible toolchains and frequent API changes make reliable multi-tool composition difficult, as small mismatches can accumulate along invocation chains, leading to brittle behavior and low scalability.

Existing approaches attempt to mitigate brittleness but remain incomplete. Some embed tool knowledge directly into model parameters (Wang et al., 2024), which reduces context demands but requires costly retraining when APIs change. Others derive static graphs from invocation logs (Liu et al., 2024b), yet sparse traces and missing parameter relations hinder generalization. Adaptive policies (Chen et al., 2024; Liu et al., 2024c) modify behavior from feedback, yet lack global structure for consistent planning. Taken together, current approaches are either structured but static or adaptive but unstructured, leaving a gap between representation and adaptation in large-scale tool reasoning. Addressing this challenge requires explicitly modeling inter-tool dependencies while maintaining resilience to API evolution. As illustrated in Figure 1, conventional agents search over disconnected tool calls (left), lacking explicit relational structure. In contrast, a structured tool graph connecting API nodes via shared parameter nodes (right) captures both dependency and composability relations, thereby enabling coherent multi-tool planning across dynamic APIs.

To realize this idea, we propose **NaviAgent**, a bilevel frame-

work that decouples high-level decision making from graph-based toolchain construction and execution. Unlike standard planner–executor formulations, the planning level does not prescribe a complete sequence of API calls in advance. Instead, it selects the next interaction mode, while the execution level dynamically constructs and revises toolchains over the tool graph. Specifically, at the planning level, NaviAgent operates in a four action decision space: direct response, intent clarification, toolchain retrieval, and tool execution. This decision space broadly covers core tool-interaction scenarios: some user intents can be resolved directly, others require clarification, and many rely on toolchain retrieval or execution. The agent focuses on selecting the appropriate action rather than reasoning over complex toolchain structures. Among these, toolchain retrieval is central to scalability, as it allows the agent to reuse and compose learned tool sequences when facing novel tasks. At the execution level, NaviAgent builds the Tool World Navigation Model (TWNM), which encodes both structural and behavioral dependencies from execution traces. By coupling these graph-based representations with navigation strategies, TWNM supports retrieval, substitution, and multi-tool composition as the ecosystem evolves. Execution feedback continually updates both TWNM and the planning policy, forming a closed loop for robust adaptation to API changes.

Comprehensive experiments on API-Bank (2k+ tools) and ToolBench (5k+ tools) show that NaviAgent achieves the highest TSR across models and task complexities while remaining efficient. We further test NaviAgent on 50 live APIs from RapidAPI, where it maintains strong performance. On DeepSeek-V3, it achieves a 12% higher TSR than ToolNet. Our main contributions are summarized as follows:

- **Bilevel Planning Framework.** We propose NaviAgent, a bilevel architecture that decouples high-level task reasoning from low-level tool execution, enabling scalable multi-tool orchestration.

- **Tool World Navigation Model (TWNM).** A unified graph-based model that captures both structural and behavioral dependencies and supports incremental integration of newly introduced or previously unseen tools in dynamic ecosystems.

- **Closed-loop evolution.** Execution feedback jointly refines the TWNM and the planning policy, forming a self-improving loop for continual adaptation to API changes.

- **Empirical Validation.** Extensive experiments on large-scale benchmarks with thousands of tools verify NaviAgent's scalability, while evaluations on 50 live APIs demonstrate its robust performance under dynamic real-world conditions.

## 2. Related Work

**Single-Tool Invocation.** Early research focused on enhancing LLMs' single-tool invocation capabilities. TALM (Parisi et al., 2022) established foundational paradigms through predefined template chains, while Easy-Tool (Yuan et al., 2024) introduced structured tool descriptions to reduce semantic parsing overhead. For long-context scenarios, tool documentation compression techniques preserved critical semantics via summarization, enabling low-resource tool usage (Xu et al., 2024). Toolformer (Schick et al., 2023) innovatively embedded tool invocation APIs in pre-training, allowing self-supervised learning of usage patterns from unlabeled data. In multimodal settings, GPT4Tools (Yang et al., 2023) improved visual tool generalization (e.g. object detection) by aligning vision-language instructions with tool descriptions.

**Multi-Tool Orchestration.** As tool libraries expanded, HuggingGPT (Shen et al., 2023) proposed a four-stage pipeline (plan, select, execute, respond) for standardized multi-tool workflows, while Chameleon (Lu et al., 2023) integrated heterogeneous tools (13+ types) via modular composition. Similarly, $\alpha$-UMI (Shen et al., 2024) decomposes the tool-use process into planning, invocation, and summarization, but uniquely assigns each stage to a dedicated lightweight LLM, enabling modular updates and improved performance, especially for smaller models. For small toolkits, TRICE (Qiao et al., 2023) optimized single tool policies via execution feedback, and ToolFactory (Ni et al., 2025) automated tool adaptation through domain-guided code synthesis. However, these approaches struggled with dynamic collaboration. For large-scale toolkits, Confucius (Gao et al., 2024) addressed combinatorial explosion via hierarchical tool classification, and ToolVerifier (Mekala et al., 2024) improved selection robustness through self-verification mechanisms.

**Dynamic Planning & Adaptation.** Static frameworks faltered under open-domain task complexity, prompting dynamic decision mechanisms. ReAct (Yao et al., 2023b) pioneered the decoupling of reasoning from tool calls through chain-of-thought planning. Building on this, Reflexion (Shinn et al., 2023) enhanced error recovery by introducing iterative self-reflection, significantly improving fault tolerance in complex tasks. For long-horizon tasks, path search techniques became pivotal: Tree-of-Thoughts (ToT) (Yao et al., 2023a) formalized tool invocation as searchable reasoning trees with dynamic branching, while ToolLLM (Qin et al., 2023) optimized search efficiency through functional hierarchy-guided DFS. ToolChain (Zhuang et al., 2023) further advanced this by employing heuristic cost estimation to prioritize high-success-rate branches. Yet, these methods assumed static tool relationships, failing to adapt to API drift

or cross-domain tasks. ControlLLM (Liu et al., 2024d) built static dependency graphs for task decomposition, whereas ToolNet (Liu et al., 2024b) Dynamic+Ally updated tool relations from historical calls, both limited by sparse multi-hop interaction data. This gap motivates our TWNM that jointly models structural dependencies and behavioral adaptations to capture evolving tool relations, aligning with findings that graph learning enhance LLM planning (Wu et al., 2024; Besta et al., 2024).

# 3. Methodology

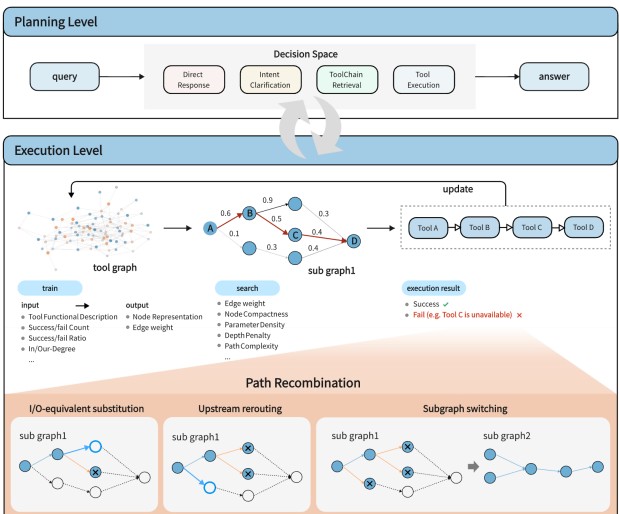

*Figure 2.* Overview of NaviAgent. It employs a bilevel framework with two interactive loops. The first loop establishes a continuous interaction between planning and execution. The second loop links the tool graph and the environment: tool execution feeds back to update the TWNM, triggering path recombination to accommodate tool changes, and inform subsequent toolchain planning. Together, these loops enable adaptive, self-improving tool reasoning under real-world API dynamics.

## 3.1. A Four-dimensional Decision Agent

### 3.1.1. DEFINITION

The architecture achieves end-to-end decision-making through LLMs, formally modeled as a quintuple $(\mathcal{H}, \mathcal{O}, \mathcal{G}, \mathcal{A}, F)$ where $\mathcal{H} = \{(o_{t-i}, a_{t-i})\}_{i=1}^{n}$ represents historical states (containing state sequence $\{o_i\}$ and action sequence $\{a_i\}$), $\mathcal{O}$ denotes the observation, $\mathcal{G}$ represents the tool dependency graph, $\mathcal{A} = \{Direct\ Response, Intent\ Clarification, ToolChain\ Retrieval, Tool\ Execution\}$ defines the four dimensional decision space, where each action corresponds to directly answering the user, requesting clarification, retrieving candidate tool dependency subgraph via graph pruning, or execute selected toolchains, respectively. $F : \mathcal{H} \times \mathcal{O} \times \mathcal{G} \to \mathcal{A}$ specifies the decision function. At each time step $t$, the agent constructs its decision context as follows. The historical context $\mathcal{H}_t$ is

defined as

$$\mathcal{H}_t = \langle (o_{t-3}, a_{t-3}), \dots, (o_{t-1}, a_{t-1}) \rangle \quad (1)$$

where a sliding window maintains the most recent three[1] observation-action pairs, capturing the agent's recent decision trajectory. The pruned tool dependency subgraph $\mathcal{G}'_{t-1} = (V, E, W)$ is computed from the agent's state at the previous time step $t - 1$, where $V$ is the node set, $E$ is the edge set, and $W$ denotes the edge weights indicating dependency strengths. The subgraph is serialized into a tree-structured textual format, ensuring a simplified yet sufficient representation for selected toolchains. The overall decision function is then formulated as

$$a_t = F(\mathcal{H}_t, \mathcal{O}_t, \mathcal{G}'_{t-1}) \quad (2)$$

where $\mathcal{O}_t$ is the current observation, and $a_t \in \mathcal{A}$ is the action selected at time $t$.

### 3.1.2. MODEL TRAINING

For supervised fine-tuning, we adopt the standard language modeling objective, computing the loss exclusively over the response or action generation segments. During training, the LLM-based agent receives as input the most recent historical state-action pairs $\mathcal{H}_t$, the current observation $\mathcal{O}_t$, and the pruned tool dependency subgraph $\mathcal{G}_{\text{sub}}$. The model is trained to maximize the likelihood of the ground-truth action $a_t^*$ at step $t$, which is derived from high-quality, curated datasets (see Appendix E.2 for details):

$$\mathcal{L}_{\text{SFT}} = -\frac{1}{N} \sum_{i=1}^{N} \log p_\theta(a_t^* \mid \mathcal{H}_t, \mathcal{O}_t, \mathcal{G}_{\text{sub}}) \quad (3)$$

where $N$ is the number of training samples and $p_\theta$ denotes the agent's predicted probability over the action space.

## 3.2. Tool World Navigation Model

### 3.2.1. GRAPH CONSTRUCTION AND REPRESENTATION

While tool standardization frameworks (e.g., Anthropic's MCP) help normalize basic API metadata, challenges remain due to inconsistent parameter naming and undocumented tool dependencies. In our framework, each tool consists of one or more APIs. We address these issues by applying semantic similarity clustering to unify functionally equivalent parameters (see details in Appendix A).

**Definition.** We construct a directed weighted graph $\mathcal{G} = (V, E, W)$ with API and parameter nodes. Edges include

---

[1]Our experiments demonstrate that utilizing the most recent three observation-action pairs achieves the best balance between accuracy and efficiency.

structural chains, defined by API schemas (e.g., parameter-to-API and API-to-parameter connections), as well as behavioral chains, derived from historical usage data (e.g., API-to-API and parameter-to-parameter dependencies). Each edge is assigned a statistical weight $\tilde{w}_{ij}$ reflecting empirical invocation patterns.

$$\tilde{w}_{ij} = \frac{N(v_i \rightarrow v_j)}{N(v_j)} \tag{4}$$

where $N(v_i \rightarrow v_j)$ counts the number of successful invocations from $v_i$ to $v_j$, and $N(v_j)$ is the total number of invocations involving $v_j$.

We formulate tool dependency discovery as a link prediction problem (Hamilton et al., 2017; Zhang & Chen, 2018; Zhou et al., 2020; Wu et al., 2024). To model this, we employ a Heterogeneous Graph Transformer (HGT) that integrates node-level feature fusion, type-specific encoding, and relation-aware message passing. Each node is initialized with both semantic (BGE-based) and structural features (including invocation statistics and degree information), and projected into a unified embedding space. We stack two multi-head HGT layers to aggregate information from the 2-hop neighborhood. Notably, the attention mechanism incorporates a statistical edge weight $\tilde{w}_{uv}$ to reflect empirical call patterns:

$$e_{uv}^{(k,r)} = \frac{(\mathbf{W}_Q^{(k,r)} \mathbf{h}_u')^\top (\mathbf{W}_K^{(k,r)} \mathbf{h}_v')}{\sqrt{d_k}} + \mathbf{b}_r^{(k)} + \tilde{w}_{uv} \tag{5}$$

$$\alpha_{uv}^{(k,r)} = \mathrm{softmax}_{u \in \mathcal{N}_r(v)} \left( e_{uv}^{(k,r)} \right) \tag{6}$$

where $\mathcal{N}_r(v)$ denotes the set of neighbors of node $v$ under relation $r$, and $\mathbf{h}_u'$, $\mathbf{h}_v'$ are the type-specific encoded representations of nodes $u$ and $v$ (see Appendix B.1 for details). $\mathbf{W}_Q^{(k,r)}$ and $\mathbf{W}_K^{(k,r)}$ are the query and key projection matrices for head $k$ and relation $r$, $\mathbf{b}_r^{(k)}$ is an edge-type-specific bias, and $d_k = d/8$ is the dimension per head. Then the concatenated head outputs are projected to obtain the final node embeddings, which are then used for link prediction.

### 3.2.2. GRAPH TRAINING OBJECTIVE

The graph model is trained with a hybrid loss that combines cross-entropy and adaptive margin objectives, both leveraging edge weights $\tilde{w}_{uv}$ to capture graded dependencies.

**Cross-entropy.**

$$\mathcal{L}_{\mathrm{CE}} = -\frac{1}{|\mathcal{E}|} \sum_{(u,v) \in \mathcal{E}} \big[ \tilde{w}_{uv} \log p_{uv} \\ + (1 - \tilde{w}_{uv}) \log(1 - p_{uv}) \big] \tag{7}$$

where $p_{uv}$ is the predicted link probability, $\tilde{w}_{uv}$ is the statistical edge weight serving as a soft label, and $\mathcal{E}$ denotes the set of all edges in the graph.

**Adaptive margin.** It assigns larger separation to higher-weight edges(i.e., $\tilde{w}_{uv} \rightarrow 1$), focusing learning on critical dependencies. For each positive edge $(u,v)^+ \in \mathcal{E}^+$, $k$ negative edges $\{(u_j, v)\}_{j=1}^k$ are sampled to construct positive and negative pairs for the margin loss.

$$m_{uv} = m_0 \left( 1 + \sigma(\tilde{w}_{uv}) \right) \tag{8}$$

$$\ell_{\mathrm{margin}}(u,v) = \frac{1}{k} \sum_{j=1}^k \big[ m_{uv} - s(u,v)^+ + s(u_j,v)^- \big]_+ \tag{9}$$

$$\mathcal{L}_{\mathrm{Margin}} = \frac{1}{|\mathcal{E}^+|} \sum_{(u,v)^+ \in \mathcal{E}^+} \ell_{\mathrm{margin}}(u,v) \tag{10}$$

where $m_0$ is a base margin, $\sigma(\cdot)$ denotes the sigmoid function, $\tilde{w}_{uv}$ is the statistical edge weight, $s(u,v)$ measures the embedding similarity, $(u,v)^+$ represents a positive edge, $(u_j,v)^-$ denotes a negative sample, and $[\cdot]_+$ is the hinge function, and $\mathcal{E}^+$ is the set of positive edges.

The final training objective is a weighted sum of the two losses:

$$\mu_t = \mu_0 \cdot \gamma^t, \ \gamma \in (0,1) \tag{11}$$

$$\mathcal{L} = \mu_t \cdot \mathcal{L}_{CE} + (1 - \mu_t) \cdot \mathcal{L}_{\mathrm{Margin}} \tag{12}$$

where $\mu_t$ is the weight for the cross-entropy loss at epoch $t$, $\mu_0$ is the initial weight, and $\gamma$ is a decay factor controlling the rate at which the contribution of the cross-entropy loss decreases over training. This curriculum strategy first emphasizes accuracy, then discrimination, yielding accurate predictions and structured embeddings.

### 3.2.3. GRAPH SEARCH

At inference time, the predicted link probabilities $p_{uv}$ are used as edge weights $w_{uv}$ in the tool graph, forming the basis for weighted-graph search and toolchain planning. We adopt two representative search strategies adapted to this setting: an Alpha-Beta Pruning method that eliminates weak toolchains using dynamic thresholds, and a heuristic search that evaluates candidate toolchains with a composite fitness balancing connectivity, depth, and cumulative weights. Complete algorithms and parameter details are provided in Appendix B.2.

### 3.2.4. GRAPH EVOLUTION

The tool ecosystem is inherently dynamic, evolving as new tools are introduced, obsolete ones are deprecated, and usage patterns shift. To systematically support such adaptability, we design three key mechanisms:

**Incremental Node Integration.** To accommodate newly introduced tools, we incrementally add new nodes via semantic similarity clustering, initializing their parameters

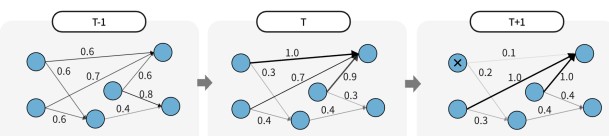

*Figure 3.* Evolution of edge weights in the TWNM. At time $T - 1$, selectable paths exhibit relatively uniform weights (similar line shades), indicating multiple comparable routes. By $T$, the TWNM has reinforced a more optimal path (the top, darker edge) through feedback learning. At $T + 1$, after the upper-left API becomes unavailable, the TWNM dynamically reconfigures the path and selects a lower route as the new optimum. This illustrates the TWNM's capability to adapt edge weights, relearn effective paths, and maintain planning flexibility in response to runtime tool changes.

(e.g., $N_{succ}(v) = 0$, $N_{fail}(v) = 0$ for successful and failed invocation counts) and the statistical weights of associated edges (e.g., $\tilde{w}_{uv} = 0$) to ensure consistency with existing graph features.

**Targeted Subgraph Pruning.** Obsolete or rarely used tools are selectively pruned based on a weighted combination of failure rates and invocation frequencies:

$$\text{Prune}(v) \propto \lambda \cdot \sigma(f_{fail}(v)) + (1 - \lambda) \cdot \sigma(f_{freq}(v)^{-1}) \quad (13)$$

where $\lambda \in [0, 1]$ controls the trade-off between failure rates and invocation frequencies, and $f_{fail}$ and $f_{freq}$ denote failure rates and invocation frequencies, respectively. Additionally, the pruning process is complemented by a dynamic reactivation step that periodically probes failed APIs and restores those that have recovered. The effectiveness of these mechanisms under changing API availability is evaluated in Appendix F.2.

**Edge Attribute Propagation.** Long-term stability and short-term adaptation are balanced by updating the statistical edge weights $\tilde{w}_{uv}$ through a combination of historical trends and recent invocation success rates:

$$\tilde{w}_{uv}^{(t)} = \eta \cdot \underbrace{\tilde{w}_{uv}^{(t-1)}}_{\text{long-term weight}} + (1 - \eta) \cdot \underbrace{\frac{N_{succ}^{\text{recent } \tau \text{ days}}(u \rightarrow v)}{N_{succ}^{\text{recent } \tau \text{ days}}(v)}}_{\text{recent success rate}}$$

$$(14)$$

where $\eta \in [0, 1]$ balances long-term memory and recent observations, and $N_{succ}^{\text{recent } \tau \text{ days}}$ denotes successful invocations within a sliding window of $\tau$ days. These dynamically updated statistical edge weights $\tilde{w}_{uv}$ are subsequently used as soft labels for supervising model training, as described in Section 3.2.2. These graph updates are performed asynchronously and therefore do not block online inference. Further details on graph-maintenance overhead are provided in Appendix E.5.

## 3.3. Dynamic Execution & Path Recombination

**NaviAgent Workflow.** When a user query arrives, NaviAgent decides whether it can respond directly, clarify the user's intent, or rely on external tools. For more complex queries, NaviAgent decomposes the task into sub-tasks and categorizes them into two types: those that can be answered or clarified immediately, and those that demand toolchain retrieval. Unlike traditional agents that fetch tools sequentially, NaviAgent searches the existing tool dependency graph for a task-relevant subgraph and selects a feasible execution path for subsequent execution. More detailed cases can be found in the Appendix C.

**Path Recombination.** If a *tool execution* action fails because an API becomes unavailable, deprecated, or malfunctioning, NaviAgent activates one of three recovery mechanisms (see the bottom of Figure 2): i) **I/O-equivalent substitution**: replacing the failed API with another that provides equivalent functionality and a compatible input–output schema, ensuring local continuity of the path; ii) **Upstream rerouting**: backtracking to the nearest upstream node and seeking an alternative subpath that can produce the required intermediate outputs; iii) **Subgraph switching**: when the current subgraph cannot reach the target, retrieving a new goal-oriented subgraph that connects to the final objective through a different route. Collectively, these strategies ensure continuous and fault-tolerant reasoning under dynamic tool conditions.

## 3.4. A Local Variational View of Mechanism Injection

Sections 3.1–3.3 describe a bilevel planning system in which the high-level agent selects among four actions, while the TWNM induces graph-conditioned execution paths together with recovery operations such as substitution, rerouting, and subgraph switching. The full policy is therefore dynamic: the admissible decisions can change after graph pruning, execution feedback, API failures, or path recombination. We do not claim that the following result fully characterizes this entire TWNM-driven process. Instead, it provides a local variational lens for one step of inference-time mechanism injection.

The abstraction is simple. For a fixed decision context, the current TWNM state specifies which actions remain admissible. Mechanism injection then acts by removing infeasible actions and preserving as much of the base policy as possible over the remaining set. The result below formalizes this as a standard information-projection statement over context-dependent feasible action sets.

**Context-dependent admissibility.** Let $\mathcal{A}$ be a finite action space and let $\pi_0(\cdot \mid h)$ be a base policy over contexts

$h \in \mathcal{H}$. For each context $h$, let

$$\mathcal{A}_{\text{feas}}(h) \subseteq \mathcal{A}$$

denote the set of actions currently admissible under the injected mechanism. Define the feasible policy class

$$\Pi_{\text{feas}} := \Big\{ \pi : \text{supp}\big(\pi(\cdot \mid h)\big) \subseteq \mathcal{A}_{\text{feas}}(h), \ \forall h \in \mathcal{H} \Big\}.$$

Under a context distribution $\mu$, consider the local KL-minimal feasible correction of the base policy:

$$\pi_{\text{inj}} \in \arg \min_{\pi \in \Pi_{\text{feas}}} \mathbb{E}_{h \sim \mu} \Big[ D_{\text{KL}}\big(\pi(\cdot \mid h) \,\|\, \pi_0(\cdot \mid h)\big) \Big]. \tag{15}$$

The next theorem shows that, within this feasible-set abstraction, the corrected policy is obtained by restricting $\pi_0$ to the admissible action set and renormalizing. Thus, the injected mechanism can be interpreted as the smallest local KL shift needed to enforce graph-conditioned feasibility.

**Theorem 3.1** (Local feasible projection). *Assume that for every context $h$ with $\mu(h) > 0$, the feasible set $\mathcal{A}_{\text{feas}}(h)$ is nonempty and*

$$\sum_{a \in \mathcal{A}_{\text{feas}}(h)} \pi_0(a \mid h) > 0.$$

*Then the optimization problem* (15) *admits a unique solution, given by*

$$\pi_{\text{inj}}(a \mid h) = \frac{\pi_0(a \mid h) \, \mathbf{1}\{a \in \mathcal{A}_{\text{feas}}(h)\}}{\sum_{a' \in \mathcal{A}_{\text{feas}}(h)} \pi_0(a' \mid h)}. \tag{16}$$

*Equivalently, in this local abstraction, mechanism injection is the information projection of the base policy onto the class of policies supported on the admissible action sets.*

**Interpretation for NaviAgent.** In our full framework, the decision context at time $t$ is

$$h_t := (H_t, O_t, G'_{t-1}),$$

and the feasible set $\mathcal{A}_{\text{feas}}(h_t)$ is induced by the current graph state and execution status. Graph pruning can rule out implausible toolchains, I/O-compatible substitution can keep only locally replaceable choices, upstream rerouting can change which intermediate actions are reachable, and subgraph switching can replace the current admissible region altogether. Theorem 3.1 does not model how these sets are generated or updated; it only describes the local correction once such a context-dependent feasible set has been specified.

**Connection to a hard-rule example.** The toy rule "if action $p$ is taken, then the next action must be $q$" is recovered as the singleton special case

$$\mathcal{A}_{\text{feas}}(h) = \begin{cases} \{q\}, & h \in H_p, \\ \mathcal{A}, & h \notin H_p. \end{cases}$$

Hence the hard-rule statement is only a degenerate one-action instance of the more general context-dependent admissibility formulation above.

**One-step correction.** The projection in (16) is idempotent: once the policy is already supported on the admissible action set at each context, applying the same projection again leaves it unchanged. This highlights the conceptual difference from parameter-updating fine-tuning: the correction is an inference-time wrapper induced by the current mechanism state, not a learned update of model weights. Proofs and auxiliary statements are deferred to Appendix I.

# 4. Experiments

## 4.1. Experimental Settings

**Datasets.** Our experiments are based on two public API benchmarks: API-Bank (Li et al., 2023) and ToolBench (Qin et al., 2023). Following the standard evaluation setup used in prior work, we construct and evaluate tasks in the simulated environments provided by these datasets, based on their API specifications, tool lists, and conversational trajectories. To validate the effectiveness of NaviAgent under real API execution conditions, we further conduct experiments on 50 live APIs sampled from the RapidAPI platform. Tasks are categorized into three levels of complexity: **Easy** (at most one API call or directly answerable), **Medium** (two API calls), and **Hard** (three or more APIs). Details of task generation are provided in Appendix D. For model fine-tuning, Qwen2.5-14B is trained on 3,500+ examples sampled from our generated task set, with strict separation between fine-tuning and evaluation data to prevent leakage.

**Baselines and Models.** The evaluation considers representative frameworks for real-world tool invocation, where handling large tool sets and enabling autonomous planning are critical. Our baselines cover diverse tool-use strategies. ReAct (Yao et al., 2023b) serves as the foundational reasoning-and-acting paradigm that interleaves reasoning with tool execution. ToolLLM (Qin et al., 2023) extends this setting with DFSDT-based planning and dynamic backtracking. ToolNet (Liu et al., 2024b) models trajectory-based relations among tools to improve tool selection and coordination, while Tool-Planner (Liu et al., 2024c) organizes tools into structured toolkits to facilitate planning in large tool spaces. We also include $\alpha$-UMI (Shen et al., 2024), which represents a multi-agent framework that decomposes tool-use workflows into modular stages coordinated by lightweight LLMs. Experiments are conducted across multiple foundation models, including open-source models (Qwen2.5-14B (Yang et al., 2024), Qwen2.5-32B (Tahmid & Sarker, 2024), DeepSeek-R1-Distill-Qwen-32B(DeepSeek-R1-32B) (Guo et al., 2025)) and closed-

| Model | Method | Easy | | | Medium | | | Hard | | | All | | |
|---|---|---|---|---|---|---|---|---|---|---|---|---|---|
| | | TCR | TSR | Steps | TCR | TSR | Steps | TCR | TSR | Steps | TCR | TSR | Steps |
| Qwen2.5-14B | ReAct | 32.4 | 26.3 | 3.52 | 24.5 | 16.8 | 3.67 | 24.8 | 20.0 | 3.64 | 27.1 | 20.6 | 3.61 |
| | ToolLLM | 56.1 | 30.4 | 4.01 | 53.8 | 19.7 | 4.02 | 38.1 | 11.4 | 4.06 | 51.0 | 21.3 | 4.03 |
| | $\alpha$-UMI | 77.7 | 39.2 | 5.53 | 77.9 | 25.0 | 5.88 | 67.6 | 13.3 | 6.07 | 75.5 | 26.9 | 5.74 |
| | ToolPlanner | 41.9 | 33.1 | 8.43 | 40.4 | 21.2 | 8.71 | 33.3 | 16.2 | 10.38 | 39.3 | 23.9 | 9.06 |
| | ToolNet | 54.7 | 40.5 | 6.12 | 50.0 | 24.0 | 6.34 | 41.9 | 18.1 | 7.49 | 49.7 | 28.0 | 6.53 |
| | **NaviAgent** | 64.2 | **50.3** | 4.18 | 60.1 | **32.3** | 4.38 | 61.1 | **22.4** | 4.68 | 61.6 | **35.8** | 4.38 |
| Qwen2.5-32B | ReAct | 33.1 | 25.0 | 3.50 | 35.6 | 24.5 | 3.60 | 30.5 | 19.0 | 3.95 | 33.6 | 23.4 | 3.63 |
| | ToolLLM | 40.5 | 31.8 | 3.67 | 48.6 | 30.3 | 3.85 | 49.5 | 23.8 | 4.10 | 46.2 | 29.3 | 3.83 |
| | $\alpha$-UMI | 78.4 | 49.3 | 5.66 | 78.8 | 26.0 | 6.02 | 77.1 | 22.9 | 6.58 | 78.3 | 32.8 | 5.94 |
| | ToolPlanner | 59.5 | 43.2 | 8.17 | 55.8 | 28.8 | 8.49 | 48.6 | 20.0 | 10.07 | 55.3 | 31.5 | 8.82 |
| | ToolNet | 70.9 | 52.7 | 5.83 | 64.4 | 30.8 | 6.11 | 57.1 | 24.8 | 7.24 | 64.9 | 36.4 | 6.27 |
| | **NaviAgent** | 88.1 | **61.1** | 4.29 | 81.7 | **41.5** | 4.60 | 79.4 | **30.8** | 5.31 | 83.2 | **45.4** | 4.66 |
| DeepSeek-V3 | ReAct | 46.6 | 36.5 | 3.52 | 58.7 | 38.5 | 3.50 | 48.6 | 23.8 | 3.74 | 52.5 | 34.5 | 3.54 |
| | ToolLLM | 56.2 | 47.4 | 3.80 | 58.8 | 30.0 | 3.92 | 29.7 | 24.8 | 3.90 | 51.3 | 34.4 | 3.86 |
| | $\alpha$-UMI | 80.8 | 59.7 | 5.95 | 89.4 | 32.9 | 5.95 | 73.0 | 29.5 | 6.64 | 82.9 | 40.7 | 6.06 |
| | ToolPlanner | 77.7 | 60.8 | 7.68 | 69.2 | 40.9 | 8.13 | 61.0 | 25.7 | 9.82 | 70.1 | 43.8 | 8.47 |
| | ToolNet | 82.4 | 62.2 | 5.41 | 77.4 | 41.8 | 5.87 | 66.7 | 26.7 | 7.03 | 76.6 | 44.9 | 6.02 |
| | **NaviAgent** | 97.9 | **71.8** | 4.40 | 96.3 | **48.5** | 4.45 | 97.0 | **44.9** | 5.19 | 97.0 | **55.2** | 4.60 |

*Table 1.* **Comparison of Baseline Frameworks on ToolBench.** TCR and TSR are reported as percentages (%), and lower Steps indicates higher efficiency. The best results are marked in **bold** and the second-best results are marked with underline.

source models (DeepSeek-V3 (Liu et al., 2024a), GPT-4o (Hurst et al., 2024)), as well as a fine-tuned lightweight model (Qwen2.5-14B).

**Metrics.** Our evaluation framework considers three metrics: task success rate (TSR), execution steps (Steps), and task completion rate (TCR). TSR and Steps are the primary indicators, with TSR measuring output quality by evaluating whether the system's response fully satisfies the user's request (via LLM-based comparison with the ground truth), and Steps reflecting execution efficiency as the total number of LLM calls required to solve a task, counted only for successfully completed tasks. TCR serves as a supplementary metric, indicating whether the system produces a final output without prematurely terminating. Tasks are considered incomplete if they exceed the maximum allowed attempts, encounter parsing errors, or fail due to input token limits. Both TCR and TSR are reported as percentages over all evaluation tasks. All experiments details of training and inference setup provided in Appendix E.2.

### 4.2. Main Results

In this section, we present the main results on ToolBench, comparing NaviAgent with strong baselines across various model sizes and task difficulties.

**Overall Performance and Efficiency.** As shown in Table 1 and Figure 4, NaviAgent consistently achieves the highest TSR across all backbone models and task difficulty levels. On the overall test set, it attains 35.8%, 45.4%, and 55.2% TSR with Qwen2.5-14B, Qwen2.5-32B, and DeepSeek-V3, outperforming the strongest baseline by 7.8,

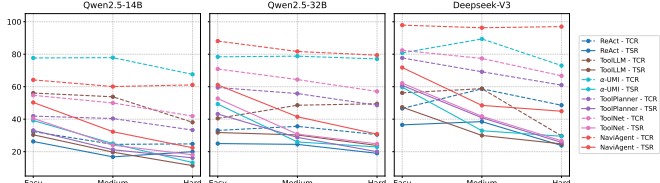

*Figure 4.* Evaluation of Frameworks on ToolBench Across Task Complexity.

9.0, and 10.3 points, respectively. The gains remain consistent across Easy, Medium, and Hard tasks, with especially clear improvements on the Hard split. Meanwhile, NaviAgent maintains moderate step counts of 4.38, 4.66, and 4.60, remaining substantially more efficient than ToolNet, ToolPlanner, and $\alpha$-UMI. Overall, NaviAgent delivers a clearly better trade-off between task success and execution efficiency.

**Relative Improvement and Robustness.** NaviAgent shows clear advantages on Hard tasks, where long-horizon planning and reliable tool selection become most critical. On the Hard split, it consistently achieves the highest TSR across all backbone models, surpassing the strongest baseline by 4.3 to 18.2 percentage points. Moreover, NaviAgent remains consistently more robust as task difficulty increases. From Easy to Hard, its TSR declines are consistently smaller than those of the strongest baselines. For example, on DeepSeek-V3, NaviAgent exhibits a relative TSR drop of 37.5%, compared with 57.1% for ToolNet and 50.6% for $\alpha$-UMI.

**Adaptability through Fine-tuning.** Notably, with supervised fine-tuning, the smaller Qwen2.5-14B model achieves performance comparable to the larger 32B model (TCR 81.2% vs 83.2%, TSR 51.3% vs 45.4%, see Figure 5 and Table 4, D+N(Heur) row), indicating that fine-tuning can effectively close the gap between model sizes. This suggests that NaviAgent remains practical in resource-constrained settings, where smaller backbones can still achieve competitive performance.

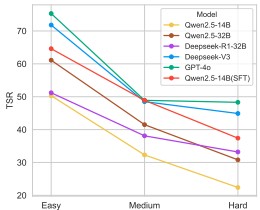

*Figure 5.* Effect of SFT on TSR.

### 4.3. Real-World Evaluation

We further evaluate NaviAgent on 50 live APIs sampled from RapidAPI to validate its practical applicability. These APIs cover seven domains: weather, air quality, restaurants, real estate, geolocation, hotels, and sports. A total of 303 user queries (101 easy, 102 medium, 100 hard) are executed through the real endpoints. As shown in Table 2, NaviAgent consistently achieves the highest TSR across all backbone models, reaching 37.4%, 54.4%, and 64.6% on Qwen2.5-14B, Qwen2.5-32B, and DeepSeek-V3, respectively, surpassing the strongest baseline by 4.3 to 12.0 percentage points. Meanwhile, NaviAgent maintains moderate execution cost, while remaining more efficient than ToolNet, ToolPlanner, and $\alpha$-UMI in both steps and runtime. These results are consistent with the simulation findings, confirming that our bilevel planning framework remains robust and effective under dynamic real-world API environments.

| Model | Method | TCR | TSR | Steps | Time |
|---|---|---|---|---|---|
| | ReAct | 32.1 | 22.1 | 3.8 | 16 |
| | ToolLLM | 53.7 | 23.8 | 4.2 | 19 |
| | $\alpha$-UMI | 77.7 | 32.4 | 6.0 | 28 |
| Qwen2.5-14B | ToolPlanner | 39.4 | 27.6 | 8.73 | 37 |
| | ToolNet | 50.8 | 33.1 | 6.41 | 31 |
| | **NaviAgent** | 65.0 | **37.4** | 5.0 | 26 |
| | ReAct | 35.4 | 25.7 | 3.9 | 19 |
| | ToolLLM | 48.7 | 30.6 | 4.0 | 23 |
| | $\alpha$-UMI | 79.4 | 42.4 | 6.2 | 33 |
| Qwen2.5-32B | ToolPlanner | 55.7 | 36.2 | 8.62 | 42 |
| | ToolNet | 66.3 | 45.1 | 6.34 | 36 |
| | **NaviAgent** | 87.6 | **54.4** | 5.0 | 28 |
| | ReAct | 55.6 | 34.1 | 3.9 | 23 |
| | ToolLLM | 53.8 | 35.7 | 4.2 | 26 |
| | $\alpha$-UMI | 85.8 | 49.3 | 6.6 | 40 |
| DeepSeek-V3 | ToolPlanner | 71.4 | 46.8 | 8.37 | 51 |
| | ToolNet | 78.2 | 52.6 | 6.07 | 44 |
| | **NaviAgent** | 99.3 | **64.6** | 5.3 | 36 |

*Table 2.* Real-World Evaluation. Time denotes the average runtime (in seconds) for completing all tasks. Other metrics are reported as in Table 1.

### 4.4. Ablation Study

**Effect of each core component.** To quantify the contribution of each design, we conduct ablations on ToolBench with DeepSeek-V3 by progressively introducing the bilevel design, tool graph, and graph search strategy. As shown in Table 3, adding the tool graph to ReAct improves TSR from 34.5% to 45.7%, showing the benefit of explicitly modeling tool relations and invocation dependencies. Using only the bilevel design also increases TSR to 42.4%, confirming its effectiveness as a key architectural design. Combining both further boosts performance from 50.3% with unpruned search to 53.4% and 55.2% with Alpha-Beta and heuristic search, respectively. Overall, these results show that the bilevel design, tool graph, and graph search strategy provide complementary gains.

| Method | TSR | Steps |
|---|---|---|
| ReAct | 34.5 | 3.54 |
| ReAct + Graph + Alpha | 45.7 | 4.21 |
| Bilevel | 42.4 | 4.78 |
| Bilevel + Graph + Unpruned | 50.3 | 5.93 |
| Bilevel + Graph + Alpha | 53.4 | 4.92 |
| Bilevel + Graph + Heuristic | **55.2** | 4.60 |

*Table 3.* Effect of each core component on ToolBench with DeepSeek-V3.

**Effect of TWNM Components.** Table 4 compares different graph construction and search variants across models. Compared with the agent-only Base, introducing graph-based planning already brings clear improvements, as evidenced by the gains of Static+A. Building on this, Dynamic+A further outperforms Static+A, with the most notable gains on hard tasks (e.g., +5.1 TSR on Qwen2.5-32B and +3.5 on GPT-4o), showing the benefit of dynamically updating the tool graph during planning. Replacing Alpha-Beta with heuristic search yields the best overall performance, bringing consistent gains of 2-3 TSR points on all tasks and around 8 points on hard cases for large models such as DeepSeek-V3 and GPT-4o. These results indicate that, beyond graph-based planning itself, both dynamic graph construction and efficient heuristic search are important for improving performance, especially on more challenging compositions. Similar trends are also observed on API-Bank (Table 9). Additional statistics on tool graph structure and link prediction are provided in Table 10.

**Behavior analysis of decision patterns.** To better understand how NaviAgent solves complex tool-use tasks, we further analyze its decision patterns in successful trajectories. As shown in Figure 6, although normal actions account for the largest proportion overall, clarification and re-retrieval are also frequently triggered, especially on more difficult tasks. This suggests that NaviAgent does not rely solely on one-pass execution, but benefits from actively resolving am-

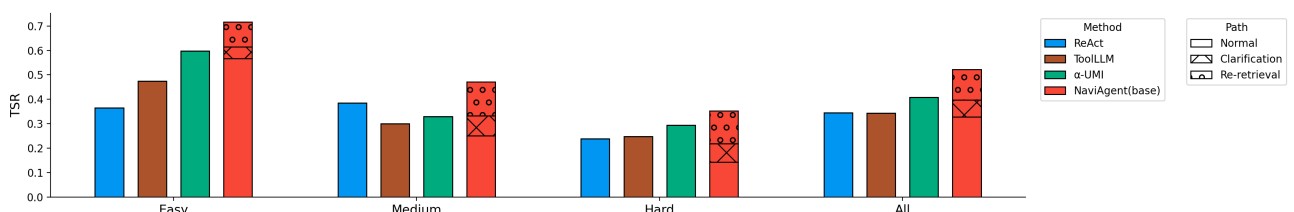

*Figure 6.* Comparison of TSR Distribution Between NaviAgent(base) and Baselines.

| Model | Method | Easy | | | Medium | | | Hard | | | All | | |
|---|---|---|---|---|---|---|---|---|---|---|---|---|---|
| | | TCR | TSR | Steps | TCR | TSR | Steps | TCR | TSR | Steps | TCR | TSR | Steps |
| Qwen2.5-14B | Base | 46.4 | 36.0 | 5.38 | 50.5 | 22.9 | 5.39 | 62.0 | 16.3 | 5.76 | 51.8 | 25.6 | 5.47 |
| | Static+A | 57.3 | 43.7 | 4.37 | 61.2 | 29.0 | 4.54 | 53.0 | 14.0 | 4.59 | 58.1 | 30.3 | 4.50 |
| | Dynamic+A | 58.8 | 48.0 | 4.31 | 61.5 | 31.7 | 4.49 | 53.3 | 16.2 | 4.61 | 58.8 | 33.4 | 4.46 |
| | **Dynamic+H** | 64.2 | **50.3** | 4.18 | 60.1 | **32.3** | 4.38 | 61.1 | **22.4** | 4.68 | 61.6 | **35.8** | 4.38 |
| Qwen2.5-32B | Base | 77.7 | 47.7 | 5.42 | 75.8 | 32.7 | 6.00 | 86.9 | 19.0 | 7.04 | 78.9 | 34.4 | 6.05 |
| | Static+A | 82.8 | 50.7 | 4.47 | 83.3 | 40.6 | 5.07 | 79.7 | 26.3 | 5.30 | 82.3 | 40.6 | 4.93 |
| | Dynamic+A | 83.1 | 51.4 | 4.41 | 85.1 | 41.3 | 5.03 | 80.0 | **31.4** | 5.37 | 83.3 | 42.3 | 4.91 |
| | **Dynamic+H** | 88.1 | **61.1** | 4.29 | 81.7 | **41.5** | 4.60 | 79.4 | 30.8 | 5.31 | 83.2 | **45.4** | 4.66 |
| Deepseek-R1-32B | Base | 89.5 | 32.2 | 6.16 | 85.0 | 25.8 | 6.64 | 88.6 | 19.7 | 6.65 | 87.3 | 26.5 | 6.49 |
| | Static+A | 92.3 | 45.8 | 5.14 | 92.5 | 35.8 | 5.39 | 91.5 | 20.8 | 5.99 | 92.2 | 35.6 | 5.45 |
| | Dynamic+A | 92.6 | **51.4** | 5.06 | 93.3 | 38.0 | 5.33 | 91.4 | 21.9 | 5.93 | 92.6 | 38.6 | 5.38 |
| | **Dynamic+H** | 93.5 | 51.2 | 4.82 | 92.4 | **38.1** | 5.23 | 87.8 | **33.2** | 5.46 | 91.7 | **41.2** | 5.15 |
| DeepSeek-V3 | Base | 93.7 | 66.3 | 5.26 | 93.8 | 39.7 | 6.00 | 94.7 | 31.1 | 6.22 | 94.0 | 46.3 | 5.81 |
| | Static+A | 92.9 | 70.5 | 4.31 | 95.8 | 47.4 | 4.66 | 93.4 | 31.1 | 5.05 | 94.3 | 51.1 | 4.64 |
| | Dynamic+A | 93.2 | 71.6 | 4.36 | 95.7 | **50.5** | 4.68 | 93.3 | 33.3 | 4.97 | 94.4 | 53.4 | 4.64 |
| | **Dynamic+H** | 97.9 | **71.8** | 4.40 | 96.3 | 48.5 | 4.45 | 97.0 | **44.9** | 5.19 | 97.0 | **55.2** | 4.60 |
| GPT-4o | Base | 92.0 | 62.7 | 5.07 | 91.0 | 35.2 | 5.67 | 94.5 | 27.8 | 6.26 | 92.1 | 42.3 | 5.61 |
| | Static+A | 99.5 | 72.1 | 4.21 | 98.3 | 43.6 | 5.35 | 97.8 | 37.9 | 5.85 | 98.6 | 51.5 | 5.10 |
| | Dynamic+A | 99.9 | **76.4** | 4.18 | 99.5 | 45.3 | 5.40 | 98.1 | 41.4 | 5.92 | 99.3 | 54.4 | 5.13 |
| | **Dynamic+H** | 99.6 | 75.3 | 4.01 | 94.5 | **48.9** | 4.71 | 98.9 | **48.3** | 5.12 | 97.1 | **57.2** | 4.58 |
| Qwen2.5-14B(SFT) | Base | 70.9 | 49.1 | 5.94 | 72.8 | 42.1 | 5.94 | 71.0 | 24.5 | 6.99 | 71.8 | 40.3 | 6.18 |
| | Static+A | 84.6 | 61.4 | 4.50 | 78.1 | 38.6 | 4.69 | 77.8 | 35.6 | 5.65 | 80.1 | 45.3 | 4.85 |
| | Dynamic+A | 85.8 | **64.9** | 4.58 | 78.4 | 39.9 | 4.75 | 78.1 | **39.0** | 5.59 | 80.7 | 47.7 | 4.89 |
| | **Dynamic+H** | 82.7 | 64.6 | 4.59 | 81.4 | **48.9** | 4.67 | 78.5 | 37.4 | 5.74 | 81.2 | **51.3** | 4.89 |

*Table 4.* **Impact of Naviagent Variants on ToolBench.** Base retains only the core agent; Static+A augments with a static graph without historical invocation data and Alpha-Beta pruning; Dynamic+A augments with a dynamic graph and Alpha-Beta pruning; Dynamic+H augments with a dynamic graph and heuristic pruning, which corresponds to our proposed NaviAgent. Metrics are reported as in Table 1. We also evaluate runtime, with detailed results reported in Appendix F.1.

biguous requests and recovering from incomplete retrieval during multi-step planning. Such a distribution also aligns with the design of NaviAgent, where explicit clarification and iterative retrieval serve as important mechanisms for handling uncertainty and compositional complexity. Additional action-level ablations in Table 11 further support this observation.

## 5. Conclusion

We presented NaviAgent, a bilevel planning framework that separates high-level decision making from low-level execution over a TWNM, achieving robust gains on ToolBench and API-Bank. In addition to benchmark evaluations, we

also conducted real-world tests, verifying NaviAgent's effectiveness and stability in practical settings. It scales to hundreds or thousands of tools with competitive efficiency and excels in complex, multi-tool tasks and larger models. Remaining challenges include handling heterogeneous tool interfaces and dynamic conditions, which may be tackled via unified protocols and adaptive graph construction. Beyond tool reasoning, NaviAgent points to broader applications: by abstracting tools as agents, its evolving graph and decision space can naturally extend to multi-agent collaboration. This perspective underscores both the challenges of building adaptive, robust systems and the opportunities for advancing toward more collaborative AI ecosystems.

## Acknowledgements

We thank the anonymous reviewers and area chair for their constructive feedback, which improved this paper. We also thank all collaborators and co-authors for their valuable discussions and contributions to this work.

## Impact Statement

This paper introduces NaviAgent, a bilevel framework for scalable tool reasoning in dynamic tool ecosystems. As function call agents become increasingly central to intelligent application development, our approach enhances their robustness to API changes and execution failures. We expect these advances to contribute to more reliable and sustainable agent systems in real-world environments.

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

# A. Graph Construction

| Original API | Original Parameter | Parameter Description | Standardized Parameter | Cluster ID |
|---|---|---|---|---|
| get_locations | name | Name of the city. | city_name | 1 |
| get_hospital_list | city | The city where the hospital is located. | city_name | 1 |
| get_hospital_list | name | Name of the hospital. | hospital_name | 2 |
| find_cheapest_prescription | city_name | The name of the city where the user wants to search for the medication. | city_name | 1 |

*Table 5.* Standardization of API Parameter

# B. Details of Graph Method

## B.1. HGT Network

This section provides a detailed description of the feature construction, network architecture, and link prediction head used in our heterogeneous graph transformer (HGT) for tool dependency modeling, supplementing the main text.

**Feature Fusion.** Each node $v$ is initialized by its semantic and structural features:

$$\mathbf{h}_v = BGE(x_v) \oplus \sigma(n_v^{succ}) \oplus \sigma(r_v^{succ}) \oplus \sigma(deg_v^{in}) \oplus \sigma(deg_v^{out}) \tag{17}$$

where $BGE(x_v)$ encodes the node description $d_v$ using BGE-Large-en-V1.5, $n_v^{succ}$ and $n_v^{fail}$ are the counts of successful and failed invocations for node $v$ (computed from historical invocation logs), $r_v^{succ} = n_v^{succ}/(n_v^{succ} + n_v^{fail})$ denotes the successful ratio, and $deg_v^{in}$ and $deg_v^{out}$ are the in-degree and out-degree of node $v$, respectively.

**Node Encoder.** To project heterogeneous nodes into a unified embedding space, we apply type-specific linear transformations, followed by non-linear activation and normalization:

$$\mathbf{h}_v' = LayerNorm \left( LeakyReLU \left( \mathbf{W}_{\tau(v)} \mathbf{h}_v + \mathbf{b}_{\tau(v)} \right) \right) \tag{18}$$

where $\mathbf{W}_{\tau(v)}$ and $\mathbf{b}_{\tau(v)}$ are the learnable weight matrix and bias for node type $\tau(v) \in \{api, param\}$, respectively.

**WeightedHGTConv Layer.** We stack two multi-head heterogeneous graph transformer (HGT) layers (each with 8 attention heads) to aggregate information from the 2-hop neighborhood. For a center node $v$ and its neighbor $u \in N_r(v)$ under edge type $r$, the attention coefficient at head $k$ is computed as:

$$\alpha_{uv}^{(k,r)} = softmax_{u \in \mathcal{N}_r(v)} \left( \frac{(\mathbf{W}_Q^{(k,r)} \mathbf{h}_u')^\top (\mathbf{W}_K^{(k,r)} \mathbf{h}_v')}{\sqrt{d_k}} + \mathbf{b}_r^{(k)} + \tilde{w}_{uv} \right) \tag{19}$$

where $\mathbf{W}_Q^{(k,r)}$ and $\mathbf{W}_K^{(k,r)}$ are the query and key projection matrices for head $k$ and relation $r$, $\mathbf{b}_r^{(k)}$ is an edge-type-specific bias, $\tilde{w}_{uv}$ is the statistical edge weight from node $u$ to $v$ (see Eq. 4, where $\tilde{w}_{ij}$ is defined for nodes $v_i$ and $v_j$), and $d_k = d/8$ is the dimension per head. The normalization $softmax_{u \in \mathcal{N}_r(v)}$ is performed over all neighbors $u$ of $v$ under relation $r$. The output embedding for node $v$:

$$\mathbf{h}_v'' = LayerNorm \left( \mathbf{h}_v' + LeakyReLU \left( \mathbf{W}_o \cdot Concat \left[ \sum_{r \in R} \sum_{u \in \mathcal{N}_r(v)} \alpha_{uv}^{(k,r)} \mathbf{W}_V^{(k,r)} \mathbf{h}_{u'} \right]_{k=1}^8 \right) \right) \tag{20}$$

where $\mathbf{W}_V^{(k,r)}$ is the value projection for head $k$ and relation $r$, $\mathbf{W}_o \in \mathbb{R}^{8d_k \times d}$ is the output projection, and $Concat[\cdot]_{k=1}^8$ denotes concatenation of outputs from all heads.

**Link Prediction.** Given the final node embeddings, the link probability between node $u$ and node $v$ is computed as:

$$p_{uv} = \sigma \left( \mathbf{W}_p \cdot Concat(\mathbf{h}''_u, \mathbf{h}''_v) + \mathbf{b} \right) \tag{21}$$

where $\mathbf{W}_p$ and $\mathbf{b}$ are learnable parameters, and $\sigma(\cdot)$ denotes the sigmoid function.

This completes the detailed description of our HGT-based network architecture.

### B.2. Graph Search Algorithm

This section provides detailed descriptions of the Alpha-Beta pruning and hybrid heuristic search algorithms, including all parameter settings, dynamic thresholding strategies, and algorithmic pseudocode.

**Alpha-Beta Pruning.** This algorithm (Knuth & Moore, 1975) is adapted for backward search over the tool dependency graph $\mathcal{G} = (V, E, W)$, parameterized by a quintuple $(\alpha, \beta, \mathcal{H}, \mathcal{D}, \mathcal{C})$, where $\alpha \in \mathbb{R}^+$ (initialized as $\alpha_0 = 0.4$) is the lower-bound threshold for acceptable path scores, and $\beta \in \mathbb{R}^+$ (with $\beta_0 = 0.9$) is the upper-bound for candidate evaluation. The dynamic threshold function $\mathcal{H}(d) = \max(0.3, 0.5 \times 0.9^d)$ applies exponential decay to balance search depth $d$ and semantic relevance. The depth attenuation factor $\mathcal{D}(d) = 1/(1 + \sqrt{d})$ penalizes longer paths. The connectivity constraint $\mathcal{C}(u, v_t) = \text{PathLength}(u, v_t) \leq 5$ ensures that generated subgraphs remain compact, where $v_t$ denotes the target node (either an API node or a parameter node). The parametric scoring function is defined as:

$$S_{uv} = \frac{w_{uv} + \mathbb{I}(u \to v_t^{\text{api}})w_{u \to v_t^{\text{api}}} + \mathbb{I}(u \to v_t^{\text{param}})w_{u \to v_t^{\text{param}}}}{3} \times \mathcal{D}(d) \tag{22}$$

where $w_{uv}$ is the direct edge weight from node $u$ to its predecessor $v$ (see Section 3.2.2), $w_{u \to v_t^{\text{api}}}$ and $w_{u \to v_t^{\text{param}}}$ denote the edge weights from $u$ to the target API node $v_t^{\text{api}}$ and target parameter node $v_t^{\text{param}}$, respectively, included only if the corresponding indicator function $\mathbb{I}(\cdot)$ is active.

During reverse depth-first search, we apply two pruning rules: Alpha-pruning is triggered at parameter nodes when $S_{uv} < \mathcal{H}(d)$ and $S_{uv} < \alpha$, while Beta-pruning is triggered at API nodes when $S_{uv} > \beta$. To further improve efficiency, the pruning thresholds are Dynamic+Ally adjusted via $\alpha' = \max(\alpha, S_{uv} \times 0.85)$ and $\beta' = \min(\beta, S_{uv} \times 1.15)$, reducing the search time complexity from $O(b^k)$ to $O((\sqrt{b})^k)$ (Knuth & Moore, 1975), where $b$ is the branching factor and $k$ is the maximum search depth. See Algorithm 1 for details.

**Heuristic Graph Search with Dynamic Pruning.** Our hybrid heuristic search algorithm combines simulated annealing (Kirkpatrick et al., 1983) and genetic algorithm strategies (Shapiro, 1999). It is parameterized by a sextuple $(\mathcal{T}_0, \eta, \mathcal{P}, d_{\max}, \mathcal{M}_\theta, \mathcal{F}_\omega)$ (see Algorithm 2), where $\mathcal{T}_0 = 200$ is the initial temperature that determines the probability of accepting suboptimal solutions and balances exploration and exploitation, $\eta = 0.7$ is the cooling rate that controls the annealing schedule $\mathcal{T}_{k+1} = \eta^{1+k/5}\mathcal{T}_k$, $\mathcal{P} = 20$ is the population size, $d_{\max} = 4$ is the maximum search depth, and $\mathcal{M}_\theta$ is a temperature-sensitive mutation operator with adaptive intensity $\theta = \lfloor \mathcal{T}/100 \rfloor$. Candidate solutions are evaluated using a composite fitness function:

$$\mathcal{F}_\omega = 0.35\mathcal{C}_c + 0.15 \log(1 + \rho_p) + 0.3\mathcal{D}_c + 0.15\mathcal{W}_n + 0.05\mathcal{C}_p \tag{23}$$

where $\mathcal{C}_c$ (node compactness) measures the closeness centrality of API nodes, $\rho_p$ (parameter density) is the ratio of parameter nodes within the subgraph to promote concise yet informative solutions, $\mathcal{D}_c = 0.2e^{-d/10} + 0.8e^{-n/8}$ (depth penalty) penalizes overly deep or complex dependency structures, with $d$ as the average depth and $n$ as the total node count, $\mathcal{W}_n$ (weight quantification) encourages solutions with higher cumulative edge weights, and $\mathcal{C}_p$ (path complexity) evaluates structural simplicity, favoring solutions with less intricate connectivity.

We parallelize the subgraph search for different target APIs in Algorithm 2. This approach processes the population evolution tasks independently and concurrently, thereby eliminating the computational bottleneck of the original algorithm's serial loops.

---

**Algorithm 1** Alpha–Beta Backward Pruning

---

1: **Input:** Graph $G$, target node $v_{\text{target}}$, $\alpha_{\text{init}} = 0.4$, $\beta_{\text{init}} = 0.9$, $d_{\text{max}} = 5$
2: **Output:** Subgraph $G_{\text{sub}}$
3: Initialize queue $Q$ with $v_{\text{target}}$; $V_{\text{visited}} = \{v_{\text{target}}\}$
4: $\alpha \leftarrow \alpha_{\text{init}}, \ \beta \leftarrow \beta_{\text{init}}$
5: **while** $Q$ not empty **do**
6:    $v \leftarrow Q.\text{pop}()$
7:    **for** each $p \in \text{predecessors}(v)$ **do**
8:      **if** $p \notin V_{\text{visited}}$ **then**
9:        $s \leftarrow \text{Score}(p \rightarrow u, \ p \rightarrow v_{\text{target}}, \ p \rightarrow v_{\text{target\_param}})$
10:       $d \leftarrow \text{current\_depth}(p)$
11:       $\mathcal{H}(d) \leftarrow \max(0.3, \ 0.5 \times 0.9^d)$
12:       **if** $p \in V_{\text{param}}$ **then**
13:         **if** $s < \mathcal{H}(d)$ and $s < \alpha$ **then**
14:           **continue**
15:         **end if**
16:         **if** $s > \beta$ **then**
17:           **break**
18:         **end if**
19:       **end if**
20:       $\alpha \leftarrow \max(\alpha, \ 0.85s)$
21:       $\beta \leftarrow \max(\beta, \ 1.15s)$
22:       $V_{\text{visited}} \leftarrow V_{\text{visited}} \cup \{p\}$
23:       $Q.\text{append}(p)$
24:      **end if**
25:    **end for**
26: **end while**
27: $V_{\text{sub}} = \{v \mid v \in V_{\text{visited}}, \ \text{PathLength}(u, v_{\text{target}}) \leq d_{\text{max}}\}$
28: **return** $G_{\text{sub}} = (V_{\text{sub}}, E) = 0$

---

## C. Case Studies

The following three cases exemplify the bilevel planning mechanism through four core actions: 1) *Direct Response*: resolves user queries using pre-trained knowledge. 2) *Intent Clarification*: initiates interactive dialogue to disambiguate vague requests. 3) *ToolChain Retrieval*: works with the TWNM to construct a pruned tool dependency subgraph, which is then returned as an executable toolchain. 4) *Tool Execution*: executes the required APIs based on the dependency subgraph, with parameter validation and state monitoring. This design achieves centralized decision control through the agent's orchestration authority while enabling dynamic resource optimization via the TWNM's graph-based toolchain generation, ensuring both efficiency and robustness of the our framework in complex task environments.

For toolchain retrieval, the request to TWNM consists of the top-3 target APIs retrieved using the LLM-generated tool description, together with the identified input parameters and desired output parameters, denoted as $R = (A_{\text{top3}}, P_{\text{in}}, P_{\text{out}})$. Given $R$, TWNM performs graph search from each recalled target API to construct candidate toolchains, using the scoring function in Appendix B.2 to guide expansion and pruning. The selected subgraph is returned to the LLM not as a raw graph, but as a serialized dependency tree as described in Sec. 3.1.1. For illustration, a simplified example is shown below:

```
TargetAPI
├── parameter_a
├── parameter_b
└── parameter_c
    └── SupportingAPI
        └── parameter_d
```

---

**Algorithm 2** Hybrid Heuristic Pruning Algorithm

---

**Require:** Dependency graph $G$, target API set $A$, initial temperature $\mathcal{T}_0 = 200$, cooling rate $\eta = 0.7$, population size $\mathcal{P} = 20$, maximum search depth $d_{\max} = 4$
**Ensure:** Optimized dependency subgraph $G^*$
 1: Initialize optimized subgraph set $S \leftarrow \emptyset$
 2: **for** each target API $a \in A$ **do**
 3:     Set temperature $\mathcal{T} \leftarrow \mathcal{T}_0$
 4:     Generate initial population $Pop$ (size $\mathcal{P}$) for API $a$
 5:     Set iteration count $k \leftarrow 0$
 6:     **while** $\mathcal{T} > 1$ and $k \leq 10$ **do**
 7:         Evaluate fitness $\mathcal{F}_\omega$ for each chromosome in $Pop$
 8:         Select elite chromosomes (top 60% based on fitness)
 9:         Generate offspring via crossover operation
10:         Apply temperature-sensitive mutation $\mathcal{M}_\theta$ with intensity $\theta = \lfloor \mathcal{T}/100 \rfloor$
11:         Update population $Pop$ with offspring
12:         Update temperature: $\mathcal{T} \leftarrow \eta^{1+k/5}\mathcal{T}$
13:         Increment iteration count $k \leftarrow k + 1$
14:     **end while**
15:     Select best chromosome from $Pop$ based on $\mathcal{F}_\omega$
16:     Decode chromosome and build API-specific subgraph $G_a$
17:     Add subgraph $G_a$ to optimized subgraph set $S$
18: **end for**
19: Merge all subgraphs in $S$ into final optimized subgraph $G^*$
20: **return** $G^* = 0$

---

## C.1. Case 1

QUERY

- Could you provide me with information about gastroenteritis? Additionally, please help me log my health data from March 5, 2025.

FIRST ROUND: INTENT ANALYSIS

1. **Gastroenteritis Inquiry**

   ACTION: **Direct Response**
   CONTEXT:

   ```
   {
   "analysis": "1. Common sense judgment: Gastroenteritis is a
   medical condition within general knowledge. 2. No API
   required for definition explanation. 3. Parameter
   verification not applicable",
   "action": "direct_answer",
   "intent": "1. Provide medical definition of gastroenteritis
   2. User's explicit request for condition explanation",
   "target_api": "",
   "params": {},
   "recall_description": "",
   "answer": "Gastroenteritis is inflammation of the stomach
   and intestines,
   typically resulting from bacterial toxins or viral
   infection. Common
   symptoms include diarrhea, vomiting, and abdominal cramps."
   ```

```
}
```

2. **Health Data Logging**

   ACTION: **TooChain Retrieval**
   CONTEXT:

```
{
"analysis": "1. Health data recording requires specialized
API 2. No matching API in candidate list 3. Critical
parameters provided:
user_id=12345, blood_pressure=130/80, heart_rate=90,
timestamp=2025-03-05",
"action": "retrieve_api",
"intent": "1. Store user health metrics 2. Provided
parameters: blood pressure, heart rate, user ID, timestamp",
"target_api": "",
"params": {},
"recall_description": "HealthDataAPI(description: Records
user health
metrics, input: user_id:string/unique identifier,
blood_pressure:string/systolic & diastolic,
heart_rate:integer/bpm,
timestamp:datetime/measurement time; output:
record_id:string/new entry identifier)",
"answer": "I need to retrieve a health data recording API to
complete this action. Should I proceed with API retrieval?"
}
```

Then, the BGE model retrieves two candidate APIs (api-RecordHealthData, api-QueryRegistration) based on the recall description above, after which the TWNM is invoked to acquire the corresponding call chains for these APIs (see Figure 7).

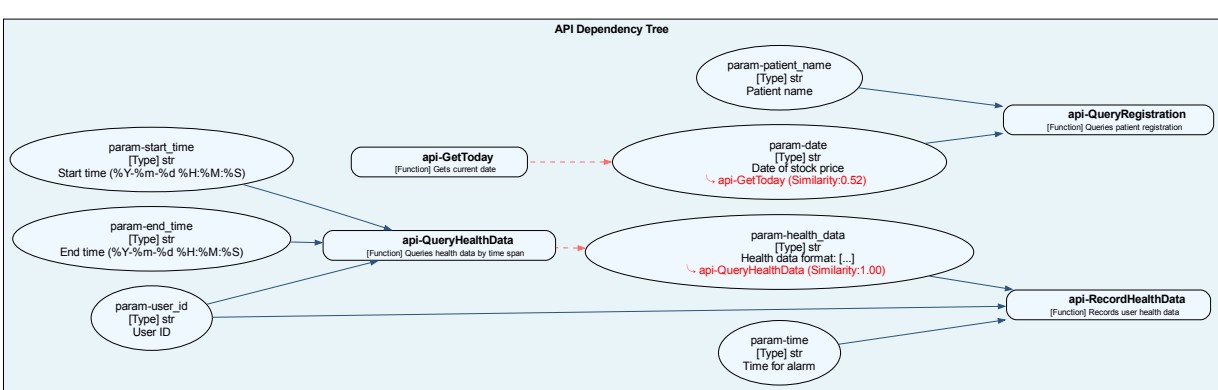

*Figure 7.* Pruned Tool Dependency Subgraph of Case1

SECOND ROUND: ITERATIVE API CHAIN EXECUTION

In this round, the agent selects and executes the optimal API chain, with api-RecordHealthData designated as the target API.

ACTION: **Tool Execution**
CONTEXT:

```
{
  "action": "call_api",
  'api_name': 'api-QueryHealthData'
  "params": {
    "user_id": "12345",
    "start_time": "2025-03-05",
    "end_time": "2025-03-05"
  }
}
```

TOOL RESPONSE:

```
{{'name':'blood_pressure','value':'130/80'}
{'name':'heart_rate','value':'90'}}
```

### THIRD ROUND: ITERATIVE API CHAIN EXECUTION

ACTION: **Tool Execution**
CONTEXT:

```
{
  "action": "call_api",
  'api_name': 'api-RecordHealthData'
  "params": {
    "user_id": "12345",
    "time": "2025-03-05 14:30:00",
    "health_data": "[{'name':'blood_pressure','value':'130/80'},
                     {'name':'heart_rate','value':'90'}]"
  }
}
```

TOOL RESPONSE:

```
Finished
```

### FOURTH ROUND: API OUTPUT VERBALIZATION

ACTION: **Direct Response**
CONTEXT:

```
{
  "analysis": "1. API parameters verified and complete
               2. Successful data storage confirmation",
  "action": "direct_answer",
  "answer": "Your health data has been successfully recorded..."
}
```

### LAST ROUND: RESPONSE SYNTHESIS

ACTION: **Direct Response**
CONTEXT:

```
1.Inflammation of gastrointestinal tract
    Common causes: Viral/bacterial infections
```

```
    Key symptoms: Diarrhea, vomiting, abdominal cramps
    Management: Hydration and rest

2. Health Data Record Confirmation:
    All parameters validated successfully through api-
    RecordHealthData. No data integrity issues detected.
```

### C.2. Case 2

QUERY

- Is Beijing in China? What is the current weather forecast for the location at latitude 40.7128 and longitude -74.0060 in metric units?

FIRST ROUND: INTENT ANALYSIS

1. **Common Sense**

   ACTION: **Direct Response**
   CONTEXT:

```
{'dependency_rank': 1,
  'intent': 'Is Beijing in China?',
  'analysis': '1. The specific problem addressed by this
  action is to determine if Beijing is located within China.
  2. The user input directly asks about the
  geographical location of Beijing. 3. This is a common
  sense question and does not require an API call. 4. No
  parameters are needed for this decision.',
  'action': 'direct_answer',
  'answer': 'Yes, Beijing is in China.'}
```

2. **Get Weather**

   ACTION: **TooChain Retrieval**
   CONTEXT:

```
{'dependency_rank': 2,
'intent': 'What is the current weather forecast for the
location at latitude 40.7128 and longitude -74.0060 in
metric units?',
'analysis': '1. The specific problem addressed by this
action is to retrieve the current weather forecast for a
specific location using latitude and longitude coordinates.
2. This is not a common sense question and requires an
external API call. 3. The API matching phase will prioritize
APIs that can provide weather forecasts based on geographic
coordinates. 4. Required parameters are latitude and
longitude, which are provided in the user input.',
'action': 'retrieve_api',
'recall_description': 'WeatherAPI(description: Retrieve
current weather conditions by geographic coordinates, input:
latitude:float/latitude coordinate;
longitude:float/longitude coordinate; output:
temperature:float/current temperature;
humidity:float/current humidity;
```

```
wind_speed:float/current wind speed)'}
```

Then, the BGE model retrieves three candidate APIs (api-current_weather_data_of_a_location_for_weather, api-by_geographic_coordinates_for_rapidweather, api-current_for_foreca_weather) based on the recall description above, after which the TWNM is invoked to acquire the corresponding call chains for these APIs (see Figure 8).

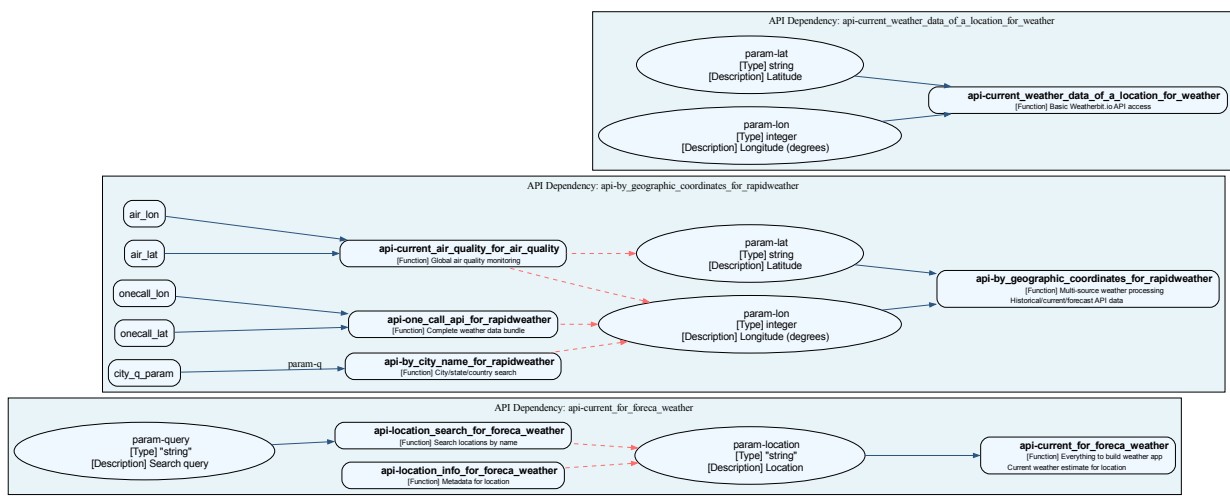

*Figure 8.* Pruned Tool Dependency Subgraph of Case2

SECOND ROUND: ITERATIVE API CHAIN EXECUTION

In this round, the agent selects and executes the optimal API chain, with api-by_geographic _coordinates_for_rapidweather designated as the target API.

ACTION: **Tool Execution**
CONTEXT:

```
{
  "action": "call_api",
  'api_name': 'api-by_geographic_coordinates_for_rapidweather'
  'params': {'param-lat': '40.7128', 'param-lon': '-74.0060'}
  }
```

TOOL RESPONSE:

```
{temperature:11.0°C, feels:8.0°C, wind_speed:5.1 m/s}
```

THIRD ROUND: API OUTPUT VERBALIZATION

ACTION: **Direct Response**
CONTEXT:

```
{
  'analysis': "1. The specific problem addressed by this
  action is to retrieve the current weather forecast for a
  specific location using the provided latitude and
  longitude. 2. The user input directly provides the
```

```
necessary parameters (latitude and longitude) and
specifies the need for metric units.
3. The target API 'api-
current_weather_data_of_a_location_for_weather' has
already been successfully called and its result is
available. 4. The required parameters (lat and lon) are
complete and the API call has been confirmed
successful.",
'action': 'direct_answer',
'answer': 'The current weather in New York is clear sky
with a temperature of 11.0°C, feels like 8.0°C, and the
wind speed is 5.1 m/s from the west-southwest.'}
```

LAST ROUND: RESPONSE SYNTHESIS

ACTION: **Direct Response**
CONTEXT:

```
Beijing is in China. The current weather in New York is
clear sky with a temperature of 11.0°C, feels ike 8.0°C, and
the wind speed is 5.1 m/s from the west-southwest.
```

**C.3. Case 3**

QUERY

- Could you tell me what the weather will be like tomorrow?

FIRST ROUND: INTENT ANALYSIS

1. **Intent Clarification**

   ACTION: **Intent Clarification**
   CONTEXT:

   ```
   {
   "analysis": "1. Weather data needs to be queried in real
   time -> not common sense\n2. Required parameter (location)
   is missing",
   "action": "clarify_intent",
   "recall_description": "",
   "answer": "Which city do you want to query tomorrow's
   weather?"
   }
   ```

2. **User Answer**

   User Answer:

   ```
   I'm in BeiJing.
   ```

SECOND ROUND: INTENT ANALYSIS

1. **Get Weather**

   ACTION: **TooChain Retrieval**
   CONTEXT:

```
{'dependency_rank': 1,
  'intent': '1. Query the weather forecast for tomorrow in
  Beijing 2. Extract location: Beijing and time: tomorrow
  from user input',
  'analysis': '1. The specific problem addressed by this
  action is to retrieve
  the weather forecast for tomorrow in Beijing. 2. The user
  input directly provides the location (Beijing) and the
  time frame (tomorrow). 3. Since no candidate APIs are
  available, the system needs to retrieve an appropriate API
  for weather forecasting. 4. There are no parameters
  provided by the user that can be used directly with an
  API, so the system must retrieve an API that can
  accept location and time as parameters.',
  'action': 'retrieve_api',
  'recall_description': 'WeatherForecastAPI(description:
  Retrieve weather forecast for a given location and date,
  input: location:string/city name;
  date:date/forecast date; output: weather:string/weather
  condition, temperature:float/forecast temperature)'}
```

Then, the BGE model retrieves three candidate APIs (api-getweatherforecast_for_apjoy_weather_forecast, api-weather_report_for_the_weather_api, api-location_info_for_foreca_weather) based on the recall description above, after which the TWNM is invoked to acquire the corresponding call chains for these APIs (see Figure 9).

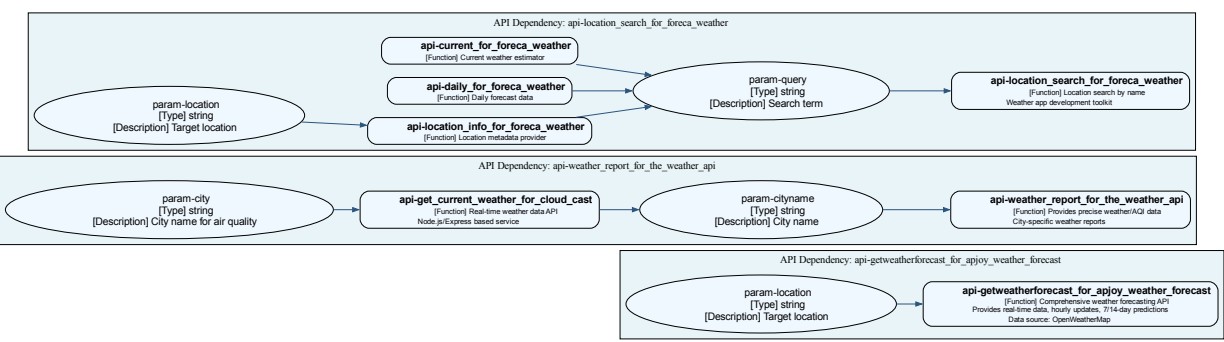

*Figure 9.* Pruned Tool Dependency Subgraph of Case3

THIRD ROUND: ITERATIVE API CHAIN EXECUTION

In this round, the agent selects and executes the optimal API chain, with api-getweatherforecast_for_apjoy_weather_forecast designated as the target API.

ACTION: **Tool Execution**
CONTEXT:

```
{
  "action": "call_api",
  'api_name': 'api-getweatherforecast_for_apjoy_weather_forecast'
  'params': {'location': 'Beijing'}
}
```

TOOL RESPONSE:

```
{temperature:25°C, humidity:60%}
```

LAST ROUND: API OUTPUT VERBALIZATION

ACTION: **Direct Response**
CONTEXT:

```
The weather forecast for tomorrow in Beijing is partly
cloudy with a temperature of 25°C and humidity of 60%.
```

## D. Data Generation

**Problem and API Dependency Chain Generation Prompts**: Automatically construct a chain of calls with strict parameter matching based on a randomful tree structure. This process involves tracing initial parameters from the leaf nodes and validating results at the root node to ensure that the generated problem aligns with the authentic API dependency logic of real-world scenarios.

```
1. Core Requirements:
   - Generate a natural-language question where:
     • Must explicitly contain initial parameters for leaf-
     node APIs
     • Implicitly requires chained API calls from leaf to
     root node
     • Root node API's output directly resolves the user's
     problem

2. Dependency Chain Rules:
   - Build parameter-passing paths where:
     • Parent API outputs must exactly match child API inputs (same
       parameter names & data types)
     • Root node API must be called last in the chain
     • The output of every leaf-node API must be utilized in
     downstream
       APIs or final results.
     • All input values must originate from either:
        Explicitly stated in the question context
        Generated by previous API outputs (no synthetic values)

3. Parameter Constraints:
   - Enforce strict value inheritance:
     • Path/query parameters must use verbatim values from:
       - User's question text
       - Preceding API response.data fields
     • Prohibit value transformation/format conversion
   - Root API output must contain realistic values matching
   its schema

4. Validation Requirements:
   - Reject generation if:
     • Missing parameter dependency between APIs
     • Input sources can't be traced to question/prior responses
     • Output fields don't fulfill next API's input requirements

5. Response Structure:
{
```

```
  "query": "<Real-world scenario requiring sequential API
  calls>",
  "answer": "<Solution derived from root API output>",
  "call_chains": [
    {
      "api_name": "<Leaf-node API>",
      "input": {
        "<param>": "<value explicitly stated in user query
        or previous API output>"
      },
      "output": {
        "status": "success",
        "data": {"<field>": "<output used by next API>"}
      }
    },
    {
      "api_name": "<Root-node API>",
      "input": {
        "<param>": "<value from previous API output>"
      },
      "output": {
        "status": "success",
        "data": {"<field>": "<realistic resolution to
        query>"}
      }
    }
  ]
}
The API dependency tree structure is as follows:
```

## E. Implementation Details

### E.1. Dataset

This table 6 system displays the sample counts of the ToolBench and API-Bank datasets, as well as the distribution of their difficulty levels.

| Dataset | Easy | Medium | Hard | Total |
|---------|------|--------|------|-------|
| API-Bank | 57 | 176 | 211 | 444 |
| ToolBench | 148 | 208 | 105 | 461 |

*Table 6.* Dataset Samples and Difficulty Distribution

### E.2. Training

We fine-tune(Tajbakhsh et al., 2016) our model using Qwen2.5-14B model with full parameter tuning. The model is trained with a maximum sequence length of 8192. We utilize a learning rate of 2e-5 and employ the AdamW optimizer with a cosine learning rate scheduler. The training process includes 10 epochs with a per-device batch size of 1 for both training and evaluation. Gradient checkpointing is enabled to reduce memory usage, and gradient accumulation is set to 4 steps to effectively manage smaller batch sizes. We apply a weight decay of 0.01 and set the maximum gradient norm to 1 for stable training. A warmup ratio of 0.03 is used to gradually increase the learning rate at the beginning of training. The training is executed on 8 Ascend 910B 64G GPUs within 10 hours. The DeepSpeed(Rasley et al., 2020) library is leveraged for efficient distributed training.

## E.3. Inference

### E.3.1. NAVIAGENT INFERENCE PROMPTS

**Inference prompts** are based on intent decomposition and dependency prioritization to achieve automatic parameter completion and error handling. They generate standardized JSON responses through hierarchical decision-making.

```
You are an intelligent API coordination system. Respond
strictly according to the following rules:

# Decision Architecture
1. **Intent Analysis**
   - Decompose compound requests into independent ordered
   sub-intents
     • Sequential dependencies first, Must execute in
     declared order
     • Parallelizable sub-intents last
     • Dependency_rank numbering for ordered execution
   - Validate parallel execution eligibility:
     • No overlapping data requirements
     • No sequential dependencies
     • Distinct parameter sets

2. **Atomic Action Formation**
     • For each validated sub-intent:
       - Create self-contained decision unit, action must
       implement full
       Decision Logic Flow
       - Maintain state separation between parallel processes
       - Focus analysis scope per sub-intent
       - Each action's analysis focuses only on its own
       intent
       - Each action analysis only solves one intent
       - Must execute each action in declared order

# Decision Logic Flow
1. **Common Sense Judgment Phase**
   - Input question -> Knowledge base matching
    Belongs to common sense -> action=direct_answer
    Requires external data -> Proceed to Phase 2

2. **API Matching Phase**
   1. If candidate_apis is empty -> action=retrieve_api
   2. Match intent with API list:
     API prioritization:
         - Complete parameters from user input
         - Minimal missing parameters
         - Shortest dependency chain
     API matching success:
       - Validate Observation in user input to confirm
       target API success:
         -> If successful -> action=direct_answer
         -> No explicit success indication:
          a) Complete parameters -> action=call_api
          (execute based on 3.1 dependency resolution)
```

```
              - If Rule 3.1c applies -> action=direct_answer
          b) Missing parameters -> Proceed to Phase 3
      API matching failed -> action=retrieve_api

3. **Parameter Completion Phase**
   - Check required parameter set:
     All parameters ready -> action=call_api
     The target API does not require parameters -> action=call_api
     Missing parameters exist:
        a) Can be completed via dependent APIs -> Execute
        Rule 3.1
        b) Use Retrieval APIs resolve parameter deficiencies
        in API
           dependencies -> action=retrieve_api
        c) Requires human confirmation -> action=clarify_intent

# Technical Rule
## 3.1 Dependency Resolution Rules
    a) Check required parameters of target API, first call
    dependent APIs.
    b) For each missing parameter, select APIs from
    dependencies not marked
       as failed.
    c) If an input parameter of an API is unavailable, use
    retrieve_api to
       call another API that generates it from known parameters.
       -> action=retrieve_api
    d) Success propagation: Completed dependency chain
       -> action=direct_answer

## 3.2 Known Failure Handling
    a) Failed APIs are recorded in failed_apis
    b) Prioritize non-failed candidate APIs

# Response Specification (Mandatory JSON Format)
[{
  "dependency_rank": 1,
  "intent": "1. <precisely describe the specific problem
  addressed by the current action>
            2. <extract data segments directly related to
            the subtask from user input>",
  "analysis":
       "<Four-level reasoning:
       1.Explicitly state the specific decision-making sub-
       intent
         addressed by this action
       2.Common sense judgment basis
       3.API matching logic (if applicable)
       4.Parameter completeness verification>",
  "action": "call_api|direct_answer|retrieve_api|clarify_intent",
  "target_api": "API name (mandatory for call_api)",
  "params": {"parameter": "value (mandatory for call_api)"},
  "recall_description":
       "When action=retrieve_api: Use 'APIName(description:
```

```
            API functionality, input: param:type/description;
            output:
            param:type/description)' format with only core
            parameters (e.g.,
            'StockAPI(description: Query stock price by symbol,
            input: symbol:string/stock symbol; output:
            price:float/current price)')",
  "answer": "When actionin[direct_answer,clarify_intent]:
  Natural language response (interrogative sentences
  required)"
}]
```

# Response Specification
Added constraint:
- JSON array items MUST be sorted by dependency_rank in
ascending order
- Sibling sub-intents should have consecutive ranks

# Termination Conditions
[OR]Generate final answer
[OR]Target API must be executed successfully, as shown in
the status

# Enforcement Constraints
1. Parameter names must strictly match API documentation
2. The 'answer' field for clarify_intent must contain
question words
3. Prioritize calling parent node APIs
4. When action in [retrieve_api]:
    - The recall_description field serves exclusively as an
    API retrieval identifier from predefined repositories.
    - parameter descriptions must distinguish between input
    and output  parameters, retaining only essential
    parameters
    - Each recall_description can only recall one
    api,multiple APIs require
      multiple actions.
5. APIs absent from Candidate APIs MUST NOT be invented
6. When action=call_api is permitted only when candidate
APIs exist and the target_api is present in the candidate
APIs.
7. The "action" field must be strictly limited to one of the
following four predefined operation types: call_api,
direct_answer,retrieve_api or clarify_intent.
8. Use retrieve_api only when:
    - Required parameters unavailable in call_api action
9. Use call_api only when:
    - The target_api is not in the list of successfully
    executed APIs
---------
# Candidate API Information:

E.3.2. INPUT GENERATION PROMPTS

**Input generation prompts**: Integrate current queries with observational data to formulate the final input, ensuring informational completeness.

```
User input:{user_input}\nPlease generate the final response
based on the following data:
{observation} :
    Requirements:
    1. Integrate all available data
    2. Indicate data limitations (if any failed APIs exist)
    3. Use natural and fluent English
```

E.3.3. API SIMULATOR PROMPTS

**API simulator prompts** are based on historical data reuse (Case1) and intelligent simulation generation (Case2/3). They achieve automated emulation of API chains through standardized JSON responses. The priority strategy is as follows: historical matching ¿ structural cloning ¿ contextual simulation.

```
Act as an API Chain Simulator to generate responses based on
historical call chains.
Follow these rules strictly:

Operation Rules:
1. Request Processing Logic
   - CASE 1: Existing API + Identical Inputs
     • Return historical outputs verbatim
     • Set {"status": "success", "type": "success"}
   - CASE 2: New API
     • Create mock data matching input format using:
       - Similar outputs from call chain (priority)
       - Simulated values (fallback)
     • Set {"status": "success", "type": "mock"}
   - CASE 3: Error
     • If not correct
     • Set {"status": "success", "type": "error"}

2. Response Requirements:
   • Strictly use JSON format only
   • Never explain parameter sources or chain structure
   • Never ask follow-up questions
   • Maintain consistent parameter naming conventions

3. Output Format (JSON):
{
  "status": "<success>", // Always 'success' per operation
  completion
  "data": <output_parameters>,
  "type": "<success/mock/error>"
}

Implementation Notes:
1. Priority Order:
   History Match > Structural Clone > Contextual Moc
```

```
API call chain is as follows:
```

E.3.4. SIMULATED USER RESPONSE AGENT PROMPTS

**Simulated user response agent prompts**: Utilize a parameter extractor as the user response to agent, serving as a simulated responder for follow-up questions by the agent. Strictly adhere to the parameter records of the API call chain to return only the queried and existent original parameter values. Automatically filter out uninvoked or null parameters to ensure that the responses include only the actual request information from the existing chain of calls.

```
As an API chain parameter extractor, directly return exact
parameter values from the given API workflows without any
modification.

## Mandatory Protocols
1. Parameter Extraction Priority
   Always return raw parameter values from the latest API
   call
   Return empty string for blank parameters (e.g. param-
   cuisines_1 -> "")

2. Response Requirements
   Merge multiple parameters in single response
    Example: "patient_id:[value] cuisine:[value]"
   Strictly avoid explanations or disclaimers
   Never reveal API structure or workflow logic

## Critical Examples
User: What's the patient ID and dietary preferences?
API Context: [param-patient_id_10:'P123' ...]
Response: patient_id:P123''

User: Current trial phase and calories limit?
API Context: [param-trial_phase_1:'Phase 2' param-
calories_max_1:'2000'...]
Response: phase:Phase 2 calories_max:2000

User: How to activate international roaming?
API Context: Relevant records
Response: I don't Know international roaming activation
information.

## Execution Context
Current API call chain:
```

## E.4. Evaluation

E.4.1. EVALUATION PROMPTS

**Evaluation prompts** in GPT-4.1 are designed to assess the correctness of the answer generation process, logical consistency, and accuracy of responses by analyzing the anticipated pathways and the decision-making pathways of the agent.

```
As an expert in response quality evaluation, you need to
perform the following steps:
I. Core Information Comparison Requirements
1. Reference Path Analysis
- Understand the simulated nature of reference API call
```

paths.
- Be aware of potential discrepancies: API names/parameter
formats may differ from actual implementations.

2. Actual Path Verification
- Compare each actual call path with the reference path.
- Focus on logical coherence rather than exact matching.

II. Error Detection Standards
1. Call Process Errors
 Parameter Anomalies:
  * Includes fictitious or illegal parameters.
 Execution Errors:
  * Returns error codes (e.g., 5xx) or invalid responses.

2. Information Integrity Errors
 Deviation in Answers:
  * Fails to address the core user query accurately.
 Missing Key Information:
  * Lacks necessary data items or explanation steps.

III. Correctness Determination Rules
1. Process Compliance
- Call sequence should be logically consistent.

2. Answer Completeness
- Covers all core elements of the user's question.
- Output provides a sufficient amount of information.

IV. Quality Rating System
[1] High-Quality Standard:
* Complete logical coherence in call paths.
* Output results are accurate and effective.
* No technical errors.

[0] Deficiency Standard (if any condition is met):
* Critical API call failures.
* Returned results do not support the answer.
* Presence of unaddressed critical errors.

V. Output Specifications
1. Detection Report Format:
    1. Parameter Validation -> Compliant/Non-compliant
    2. Path Verification -> Compliant/Non-compliant
    3. Result Completeness -> Compliant/Non-compliant

2. Final Conclusion Format:
{'Quality Result': 1} or {'Quality Result': 0}

VI. Input Data Interface
User Question: {question}
[AGENT Answer Start]
{reference}
[AGENT Answer End]

```
[Reference Call Path]
{reference_chain}
[Reference Call Path End]
[Actual Call]
{agent_actual_chain}
[Actual Call End]
```

### E.5. Graph Maintenance overhead

All maintenance measurements were obtained independently of inference runtime, and each operation runs asynchronously without affecting query latency. Specifically, graph maintenance involves three types of operations: (1) Node attributes are updated asynchronously in real time upon each successful or failed tool invocation. These updates are lightweight (55 ms per event) and modify only local node attributes without reloading the entire graph. (2) New tool insertions are also handled asynchronously in real time, taking under 100 ms per tool including semantic matching and attribute initialization; such events occur infrequently compared with the total number of queries. (3) Full graph retraining is periodically performed offline (94 s on a single NVIDIA P40) on a separate copy and asynchronously hot-swapped into production, ensuring continuous availability.

## F. Experiments

### F.1. Runtime Experiments

| Model | Method | Easy | Medium | Hard | All |
|---|---|---|---|---|---|
| Qwen2.5-14B | Base | 26.6 | 34.0 | 44.5 | 34.0 |
| | Static+A | 21.6 | 27.1 | 36.1 | 27.4 |
| | Dynamic+A | 19.8 | 25.3 | 34.4 | 25.6 |
| | **Dynamic+H** | 22.9 | 30.4 | 39.6 | 30.1 |
| Qwen2.5-32B | Base | 33.4 | 41.7 | 53.3 | 41.7 |
| | Static+A | 24.0 | 33.0 | 41.1 | 32.0 |
| | Dynamic+A | 24.0 | 31.0 | 38.6 | 30.5 |
| | **Dynamic+H** | 28.1 | 33.8 | 48.4 | 35.3 |
| Deepseek-R1-32B | Base | 36.2 | 44.2 | 61.2 | 45.5 |
| | Static+A | 27.5 | 33.7 | 46.8 | 34.7 |
| | Dynamic+A | 24.6 | 34.6 | 44.5 | 33.6 |
| | **Dynamic+H** | 31.0 | 36.7 | 49.7 | 37.8 |
| DeepSeek-V3 | Base | 43.6 | 56.6 | 71.3 | 55.8 |
| | Static+A | 34.4 | 44.5 | 55.5 | 43.8 |
| | Dynamic+A | 30.3 | 41.6 | 53.3 | 40.6 |
| | **Dynamic+H** | 37.0 | 47.3 | 61.6 | 47.3 |
| GPT-4o | Base | 42.3 | 55.6 | 75.6 | 55.9 |
| | Static+A | 34.9 | 44.1 | 59.8 | 44.7 |
| | Dynamic+A | 32.6 | 43.8 | 56.4 | 43.1 |
| | **Dynamic+H** | 36.9 | 47.1 | 61.5 | 47.1 |
| Qwen2.5-14B(SFT) | Base | 27.0 | 38.0 | 50.1 | 37.2 |
| | Static+A | 22.3 | 27.1 | 37.8 | 28.0 |
| | Dynamic+A | 19.9 | 27.9 | 35.9 | 27.2 |
| | **Dynamic+H** | 24.5 | 31.4 | 40.6 | 31.3 |

*Table 7.* Runtime(in Seconds) of NaviAgent Variants on ToolBench

Table 7 presents the runtime (in seconds) of NaviAgent variants across different models on ToolBench. Notably, the Dynamic+A method consistently achieves lower runtime across all models, with the most significant improvement observed in DeepSeek-V3: compared to the Base method (55.8 seconds), Dynamic+A reduces the runtime by 15 seconds, corresponding to a relative improvement of approximately 26.9%. Among all methodological variants, Dynamic+H demonstrates the optimal overall performance; however, it is constrained by higher runtime induced by heuristic strategies and excessive

search scale, which will be the focus of subsequent optimization efforts.

## F.2. Pruning and Reactivation

To evaluate the pruning and reactivation mechanisms under changing API availability, we sampled 50 queries from ToolBench and constructed a two-phase setting on the same query set. In Phase 1, 10% of APIs were randomly designated as unavailable upon invocation. In Phase 2, these APIs were assumed to recover. After each query, we updated API failure and usage statistics, pruned APIs whose pruning scores in Eq. 13 exceeded 0.7, and randomly selected 10% of the pruned APIs for reactivation. We compared two settings: (1) NaviAgent without pruning/reactivation and (2) the full NaviAgent with dynamic graph maintenance enabled. Table 8 shows that dynamic pruning and reactivation improve both task success rate and efficiency under changing API availability. Compared with the variant without these mechanisms, the full NaviAgent increases TSR from 44.0% to 48.0% while reducing the average number of steps from 5.12 to 4.72. These results show that pruning reduces wasted exploration over failing APIs, while reactivation helps recover useful tools after availability changes.

| Method | TSR (%) | Steps |
|---|---|---|
| w/o mechanisms | 44.0 | 5.12 |
| NaviAgent | 48.0 | 4.72 |

*Table 8.* Evaluation of pruning and reactivation.

## F.3. Experiments on API-Bank

Table 9 demonstrates that the experimental outcomes of the API-Bank dataset are consistent with those observed in the ToolBench-based experiments.

| Model | Method | Easy | | | Medium | | | Hard | | | All | | |
|---|---|---|---|---|---|---|---|---|---|---|---|---|---|
| | | TCR | TSR | Steps | TCR | TSR | Steps | TCR | TSR | Steps | TCR | TSR | Steps |
| Qwen2.5-14B | Base | 47.8 | 33.4 | 5.40 | 60.5 | 24.8 | 6.09 | 71.6 | 29.9 | 6.47 | 63.1 | 27.6 | 6.06 |
| | Static+A | 63.4 | 44.8 | 4.88 | 72.4 | 32.8 | 5.38 | 68.4 | 34.3 | 5.41 | 67.9 | 34.0 | 5.22 |
| | Dynamic+A | 64.9 | 49.1 | 4.93 | 72.7 | 36.4 | 5.32 | 68.7 | 36.0 | 5.36 | 68.3 | 36.7 | 5.18 |
| | **Dynamic+H** | 73.1 | 56.1 | 4.71 | 66.6 | 40.4 | 5.27 | 66.4 | 33.5 | 5.63 | 65.7 | 37.9 | 5.26 |
| Qwen2.5-32B | Base | 61.6 | 46.6 | 5.26 | 78.6 | 35.0 | 6.84 | 68.9 | 30.7 | 7.38 | 70.4 | 33.4 | 6.78 |
| | Static+A | 88.6 | 65.7 | 4.63 | 80.0 | 34.3 | 5.90 | 84.6 | 39.9 | 6.29 | 81.3 | 39.5 | 5.82 |
| | Dynamic+A | 89.5 | 68.4 | 4.57 | 80.1 | 35.8 | 5.86 | 85.3 | 45.0 | 6.36 | 81.8 | 42.8 | 5.83 |
| | **Dynamic+H** | 90.8 | 74.0 | 4.54 | 87.0 | 44.7 | 5.70 | 84.2 | 45.5 | 5.66 | 84.1 | 47.2 | 5.43 |
| Deepseek-R1-32B | Base | 88.2 | 66.2 | 6.41 | 65.3 | 28.3 | 7.79 | 64.5 | 25.4 | 7.83 | 65.9 | 30.3 | 7.49 |
| | Static+A | 89.2 | 60.7 | 5.77 | 88.1 | 46.1 | 6.83 | 81.2 | 30.5 | 6.82 | 83.0 | 39.2 | 6.56 |
| | Dynamic+A | 89.5 | 63.2 | 5.73 | 90.9 | 48.3 | 6.75 | 81.5 | 34.1 | 6.76 | 84.2 | 42.0 | 6.49 |
| | **Dynamic+H** | 99.1 | 77.6 | 4.96 | 89.4 | 46.9 | 6.06 | 79.3 | 34.4 | 6.81 | 83.6 | 43.2 | 6.16 |
| DeepSeek-V3 | Base | 86.3 | 67.0 | 5.95 | 86.3 | 46.5 | 6.65 | 85.4 | 42.1 | 7.09 | 83.9 | 45.5 | 6.64 |
| | Static+A | 97.7 | 77.4 | 4.84 | 98.6 | 55.5 | 6.17 | 98.8 | 48.1 | 5.82 | 96.4 | 53.1 | 5.72 |
| | Dynamic+A | 99.9 | 82.5 | 4.77 | 98.9 | 58.5 | 6.21 | 99.1 | 51.2 | 5.87 | 96.9 | 56.3 | 5.76 |
| | **Dynamic+H** | 98.8 | 88.9 | 5.00 | 98.0 | 60.0 | 5.74 | 98.6 | 52.3 | 5.88 | 96.2 | 58.0 | 5.60 |
| GPT-4o | Base | 96.8 | 74.9 | 5.23 | 92.4 | 48.3 | 6.15 | 94.5 | 38.6 | 6.19 | 91.8 | 45.4 | 5.93 |
| | Static+A | 99.6 | 76.8 | 4.15 | 98.6 | 54.5 | 5.17 | 98.3 | 46.9 | 4.85 | 96.3 | 52.0 | 4.79 |
| | Dynamic+A | 99.9 | 78.5 | 4.14 | 98.9 | 56.4 | 5.14 | 98.6 | 52.2 | 4.90 | 96.6 | 55.5 | 4.80 |
| | **Dynamic+H** | 98.9 | 76.1 | 3.70 | 98.1 | 57.9 | 5.00 | 97.0 | 57.8 | 5.00 | 95.5 | 58.5 | 4.75 |
| Qwen2.5-14B(SFT) | Base | 76.0 | 45.4 | 5.63 | 74.9 | 35.1 | 6.35 | 76.3 | 40.6 | 6.39 | 74.0 | 38.0 | 6.15 |
| | Static+A | 94.1 | 60.6 | 4.69 | 88.7 | 41.3 | 5.24 | 87.9 | 41.3 | 5.28 | 86.9 | 42.4 | 5.08 |
| | Dynamic+A | 94.7 | 64.3 | 4.67 | 89.8 | 44.1 | 5.32 | 88.2 | 42.2 | 5.34 | 87.5 | 44.3 | 5.14 |
| | **Dynamic+H** | 93.2 | 71.0 | 4.61 | 90.2 | 48.3 | 5.17 | 87.6 | 44.5 | 5.14 | 87.3 | 47.8 | 4.98 |

*Table 9.* **Impact of NaviAgent Variants on API-Bank.** Metrics are reported as in Table 4.

| Dataset | APIs | Nodes | Edges | ACC | F1 | AUC |
|---------|------|-------|-------|-----|-----|-----|
| ToolBench | 5501 | 7866 | 24215 | 76.4 | 77.6 | 0.75 |
| API-Bank | 2650 | 6025 | 10255 | 78.4 | 76.1 | 0.71 |

*Table 10.* **Tool Graph Statistics and Link Prediction Evaluation.** Nodes and Edges denote the number of nodes and edges in the graph, respectively. ACC and F1 are reported as percentages (%), while AUC is reported as a value between 0 and 1.

## G. Link Prediction Evaluation

## H. Action-level ablation.

We further examine the effect of several decision-level designs in NaviAgent on ToolBench with DeepSeek-V3. Removing clarification decreases TSR from 53.4% to 47.6%, indicating that explicit clarification helps resolve ambiguous user requests. Merging retrieval and execution into a single action also lowers TSR to 50.8%, suggesting that decoupling tool retrieval from execution leads to more reliable planning. These results further support the role of explicit decision mechanisms in complex tool-use tasks.

| Method | TSR | Steps |
|--------|-----|-------|
| w/o Clarification | 47.6 | 4.68 |
| w/ Merged Retrieval-Execution | 50.8 | 4.51 |
| NaviAgent | 53.4 | 4.62 |

*Table 11.* Action-level ablation on ToolBench with DeepSeek-V3.

## I. A Local Variational View of Mechanism Injection

This appendix gives the details behind the local variational interpretation in Section 3.4. The goal is deliberately modest. We do not attempt to prove that the entire TWNM-driven dynamic policy is the solution of a fixed global optimization problem, since admissible actions may change after graph evolution, retrieval outcomes, runtime failures, and path recombination. Instead, we isolate a context-dependent feasibility model for one local decision step and show that the resulting injected policy is the KL-minimal correction of a base policy.

### I.1. Setup: contexts, actions, and feasible sets

Let $\mathcal{A} = \{1, 2, \ldots, d\}$ be a finite action space, and let $\mathcal{H}$ denote the set of decision contexts. A stochastic policy is a map

$$\pi(\cdot \mid h) \in \Delta(\mathcal{A}), \qquad h \in \mathcal{H}.$$

We fix a reference (base) policy $\pi_0$ and an arbitrary probability distribution $\mu$ over contexts.

To model mechanism injection, we assume that each context $h$ is associated with a nonempty feasible action set

$$\mathcal{A}_{\text{feas}}(h) \subseteq \mathcal{A}.$$

The interpretation is that $\mathcal{A}_{\text{feas}}(h)$ collects the actions that remain admissible after incorporating the current mechanism state. In NaviAgent, this set may be induced by the current history, observation, graph state, graph pruning result, or recovery operation; for the local result below, we only require that it is specified as a function of the context.

We define the feasible policy class by

$$\Pi_{\text{feas}} := \Big\{ \pi : \text{supp}\big(\pi(\cdot \mid h)\big) \subseteq \mathcal{A}_{\text{feas}}(h), \ \forall h \in \mathcal{H} \Big\}. \tag{24}$$

### I.2. KL-minimal feasible correction

For a fixed feasible-set map, we model inference-time correction as the KL-minimal modification of the base policy subject to the feasibility constraint:

$$\pi^{\star} \in \arg \min_{\pi \in \Pi_{\text{feas}}} \mathbb{E}_{h \sim \mu} \Big[ D_{\text{KL}}\big(\pi(\cdot \mid h) \,\|\, \pi_0(\cdot \mid h)\big) \Big]. \tag{25}$$

We impose the following mild nondegeneracy condition.

**Assumption I.1** (Support on feasible actions)**.** For every context $h$ with $\mu(h) > 0$, the feasible set $\mathcal{A}_{\text{feas}}(h)$ is nonempty and

$$Z(h) := \sum_{a \in \mathcal{A}_{\text{feas}}(h)} \pi_0(a \mid h) > 0.$$

Assumption I.1 ensures that the KL objective in (25) is finite for at least one feasible policy. Under this condition, the optimizer has a closed form.

**Theorem I.2** (Local KL projection onto context-dependent feasible sets)**.** *Under Assumption I.1, problem (25) admits a unique solution $\pi^\star$, given by*

$$\pi^\star(a \mid h) = \frac{\pi_0(a \mid h)\mathbf{1}\{a \in \mathcal{A}_{\text{feas}}(h)\}}{Z(h)}, \qquad Z(h) := \sum_{a' \in \mathcal{A}_{\text{feas}}(h)} \pi_0(a' \mid h). \tag{26}$$

*Equivalently, $\pi^\star$ is obtained by restricting the base policy to the feasible action set and renormalizing. This is a local information projection, not a full characterization of the global TWNM update process.*

*Proof.* Because the objective in (25) is an expectation over $h$, the optimization separates pointwise across contexts. Thus it suffices to fix $h$ and solve

$$\min_{p \in \Delta(\mathcal{A}):\, \text{supp}(p) \subseteq \mathcal{A}_{\text{feas}}(h)} D_{\text{KL}}\big(p \,\|\, \pi_0(\cdot \mid h)\big). \tag{27}$$

Let $S := \mathcal{A}_{\text{feas}}(h)$ and write $\pi_0^h(a) := \pi_0(a \mid h)$. Any feasible $p$ satisfies $p(a) = 0$ for $a \notin S$, so

$$D_{\text{KL}}(p\|\pi_0^h) = \sum_{a \in S} p(a) \log \frac{p(a)}{\pi_0^h(a)}.$$

Add and subtract $\log Z(h)$ inside the logarithm:

$$\sum_{a \in S} p(a) \log \frac{p(a)}{\pi_0^h(a)} = \sum_{a \in S} p(a) \log \frac{p(a)}{\pi_0^h(a)/Z(h)} - \log Z(h).$$

Define

$$\widetilde{\pi}^h(a) := \frac{\pi_0^h(a)\mathbf{1}\{a \in S\}}{Z(h)}.$$

Then $\widetilde{\pi}^h \in \Delta(S)$ by Assumption I.1, and the previous display becomes

$$D_{\text{KL}}(p\|\pi_0^h) = D_{\text{KL}}(p\|\widetilde{\pi}^h) - \log Z(h).$$

Since $D_{\text{KL}}(p\|\widetilde{\pi}^h) \geq 0$ with equality iff $p = \widetilde{\pi}^h$, the unique minimizer of (27) is $p = \widetilde{\pi}^h$. Applying this argument for each $h$ yields (26). Uniqueness follows from the pointwise uniqueness. $\square$

### I.3. Idempotence of the feasible projection

The projection defined in Theorem I.2 is idempotent.

**Proposition I.3** (Idempotence)**.** *Let $\mathcal{P}_{\text{feas}}$ denote the operator that maps a policy $\pi$ to its feasible-set projection,*

$$(\mathcal{P}_{\text{feas}}\pi)(a \mid h) = \frac{\pi(a \mid h)\mathbf{1}\{a \in \mathcal{A}_{\text{feas}}(h)\}}{\sum_{a' \in \mathcal{A}_{\text{feas}}(h)} \pi(a' \mid h)},$$

*whenever the denominator is positive. Then*

$$\mathcal{P}_{\text{feas}}(\mathcal{P}_{\text{feas}}\pi) = \mathcal{P}_{\text{feas}}\pi.$$

*In particular, once a policy is already supported on the feasible set at every context, applying the same correction again produces no further change.*

*Proof.* Let $\widehat{\pi} := \mathcal{P}_{\text{feas}}\pi$. By construction, for every $h$,

$$\text{supp}\big(\widehat{\pi}(\cdot \mid h)\big) \subseteq \mathcal{A}_{\text{feas}}(h).$$

Hence

$$\sum_{a' \in \mathcal{A}_{\text{feas}}(h)} \widehat{\pi}(a' \mid h) = 1,$$

and therefore

$$(\mathcal{P}_{\text{feas}}\widehat{\pi})(a \mid h) = \frac{\widehat{\pi}(a \mid h)\mathbf{1}\{a \in \mathcal{A}_{\text{feas}}(h)\}}{1} = \widehat{\pi}(a \mid h).$$

This proves the claim. $\qquad\square$

### I.4. Graph-conditioned interpretation for NaviAgent

The preceding formulation connects to NaviAgent by taking the context to include the current execution state. At time $t$, one may interpret

$$h_t := (H_t, O_t, G'_{t-1}),$$

where $H_t$ is the recent interaction history, $O_t$ is the current observation, and $G'_{t-1}$ is the current pruned subgraph. The feasible set $\mathcal{A}_{\text{feas}}(h_t)$ then represents the actions currently admissible under the TWNM state.

This viewpoint remains intentionally abstract. In the full system, the feasible set may itself be generated by graph pruning, I/O-equivalent substitution, upstream rerouting, or subgraph switching, and may therefore vary over time. Theorem I.2 does not model how those sets are produced or how the graph evolves; it identifies the KL-minimal local correction once the admissible action set for the current context has been specified.

### I.5. Hard-rule result as a singleton special case

We now recover an "if $p$, then $q$" rule as a corollary of the general feasible-set theorem. This shows that the hard-rule example is only a singleton special case, whereas the feasible-set formulation allows the admissible set to vary with the full context.

Fix two distinguished actions $p, q \in \mathcal{A}$, and let $H_p \subseteq \mathcal{H}$ denote the subset of contexts in which the previous action encoded in $h$ equals $p$. Define

$$\mathcal{A}_{\text{feas}}(h) = \begin{cases} \{q\}, & h \in H_p, \\ \mathcal{A}, & h \notin H_p. \end{cases}$$

Then the feasible class in (24) becomes

$$\Pi_{\text{mech}} = \{\pi : \forall h \in H_p, \ \pi(q \mid h) = 1\},$$

and the projected policy takes the form

$$\pi_{\text{mech}}(\cdot \mid h) = \begin{cases} \delta_q(\cdot), & h \in H_p, \\ \pi_0(\cdot \mid h), & h \notin H_p, \end{cases}$$

where $\delta_q$ is the Dirac distribution at action $q$.

**Corollary I.4** (Hard local rule as KL information projection). *Assume that $\pi_0(q \mid h) > 0$ whenever $h \in H_p$ and $\mu(h) > 0$. Then $\pi_{\text{mech}}$ is the unique solution of*

$$\min_{\pi \in \Pi_{\text{mech}}} \mathbb{E}_{h \sim \mu}\Big[D_{\text{KL}}\big(\pi(\cdot \mid h) \,\|\, \pi_0(\cdot \mid h)\big)\Big].$$

*Equivalently, the hard rule "if the previous action is $p$, then the next action must be $q$" is the singleton special case of the context-dependent feasible-set projection.*

*Proof.* For $h \in H_p$, the feasible set is $\{q\}$, so (26) gives

$$\pi^\star(a \mid h) = \mathbf{1}\{a = q\} = \delta_q(a).$$

For $h \notin H_p$, the feasible set is all of $\mathcal{A}$, so the projection leaves $\pi_0(\cdot \mid h)$ unchanged. This is exactly $\pi_{\text{mech}}$. $\qquad\square$

## I.6. Remarks

**Why this appendix is intentionally modest.**    The value of the result is interpretive rather than comprehensive: it does not claim that the full TWNM dynamics reduce to a fixed convex projection problem. Instead, it isolates a simple and transparent variational principle that clarifies one aspect of mechanism injection: once an admissible decision region is specified by the current graph-conditioned mechanism, the corresponding inference-time correction is the smallest KL shift from the base policy that enforces that region.

**Relation to inference-time control.**    The projection formula (26) can be viewed as a gradient-free policy wrapper. Unlike fine-tuning, it does not modify model parameters; instead, it edits the support of the decision rule at inference time using the current feasible set and renormalizes the remaining probability mass.

**A possible soft extension.**    A natural extension is to replace hard feasibility by a context-dependent mechanism score $r(h, a)$ and solve a KL-regularized reweighting problem. The resulting optimizer has the Gibbs form

$$\pi^{\star}(a \mid h) \propto \pi_0(a \mid h) \exp\big(\tau r(h, a)\big),$$

while the hard feasible-set projection is recovered as the limiting case $r(h, a) \in \{0, -\infty\}$. We do not pursue this extension here, since the main purpose of this appendix is to provide a minimal interpretive abstraction rather than a full control-theoretic model.

