# OpenReview forum: "NaviAgent: Graph‑Driven Bilevel Planning for Scalable Tool Orchestration"
_ICML.cc/2026/Conference — ICML 2026 regular_

### Official Review · Reviewer_pvXC · 2026-03-07

**Soundness:** 4
**Presentation:** 3
**Significance:** 4
**Originality:** 3
**Overall Recommendation:** 5
**Confidence:** 3

**Summary:**

This paper proposes NaviAgent, a bilevel framework for multi-tool orchestration that decouples high-level task planning (handled by an LLM) from low-level tool execution (handled by a graph-based Tool World Navigation Model, or TWNM). The TWNM encodes structural and behavioral dependencies among tools as a directed weighted graph, trained via link prediction to generalize beyond observed co-occurrences. Evaluations on API-Bank, ToolBench, and 50 live RapidAPI endpoints show consistent improvements in task success rate over baselines, with thorough ablations confirming the contribution of each component.

**Compliance With Llm Reviewing Policy:**

Affirmed.

**Final Justification:**

My final recommendation for this work is 5 Accept. It's strong for the reasons outlined above, and I feel that the clarifications/revisions the authors proposed in the rebuttal period addressed many of my remaining questions and will make the final paper stronger.

**Key Questions For Authors:**

1. What is the precise interface between the LLM and the tool graph? When the LLM selects "ToolChain Retrieval," what is the format of the request to the graph search module, and how are results presented back to the LLM for execution?
2. Can the pruning/reactivation mechanism be evaluated, either via failure simulation on the benchmarks or via analysis of its real-world usage in the RapidAPI experiments?
3. What does "real-time API execution is currently unavailable" mean in Section 4.1, and what are the implications for the validity of the simulated evaluation?

**Limitations:**

The authors have adequately discussed potential limitations of this work

**Strengths And Weaknesses:**

### Strengths

**Strong motivation.** The problem of navigating complex, interdependent toolchains under real-world dynamics (API failures, deprecation, evolving ecosystems) is well-motivated and practically important.

**Sound and interesting graph-based tool modeling.** The TWNM is a nice bridge between the fuzzy query understanding capabilities of LLMs and more formal, robust reasoning over tools and APIs. The combination of structural edges (from schemas) and behavioral edges (from invocation logs), trained via a heterogeneous graph transformer, is well-designed.

**Strong empirical and theoretical support.** The tool graph component is justified both theoretically (Section 3.4) and with extensive, convincing experiments. The ablation study (Table 3) clearly isolates the value of dynamic graphs and heuristic search, and the real-world RapidAPI evaluation adds credibility.

### Weaknesses

**Underspecified LLM–graph interface.** The interface between the LLM planner and the TWNM is the architectural centerpiece of this framework, yet it is poorly explained. When the LLM selects "ToolChain Retrieval," how is the natural language intent translated into a graph query? What is the type signature of that request? How are candidate subgraphs selected and presented back to the LLM? Section 3.3's "NaviAgent Workflow" paragraph is too brief to answer these questions, and Appendix C contains verbose examples that do not clarify the high-level mechanism. This is a critical gap, because understanding this interface is essential for assessing whether the system would be practical to deploy.

**No evaluation of pruning/reactivation mechanisms.** The targeted subgraph pruning (Eq. 13) and dynamic reactivation of recovered APIs are presented as key features for handling real-world API dynamics, but they are never meaningfully evaluated. The ToolBench/API-Bank experiments are static and simulated. The RapidAPI evaluation would be the natural place to test these mechanisms, either through failure simulation or by reporting how often pruning/reactivation was actually triggered, but no such analysis is provided.

### Minor concerns

**Unclear statement about real-time execution.** Section 4.1 states "as real-time API execution is currently unavailable" without explanation. It's unclear whether this refers to a platform limitation, cost constraint, or something else.

**Fine-tuning necessity is unclear from the main text.** The paper presents SFT (Section 3.1.2) before the main results, which implies it is a core component. In practice, the main results (Table 1) are all zero-shot, and SFT is only used to boost the smaller 14B model. This ordering is somewhat misleading and could be reorganized for clarity.

---

> ### Author Rebuttal · Authors · 2026-03-31
>
> **Q1: What is the type signature of the LLM–graph interface? How are candidate subgraphs selected and presented back to the LLM?**
>
> A1: The request to TWNM consists of three elements: (1) the top-3 target APIs retrieved using the LLM-generated tool description as the retrieval query, (2) the input parameters identified from the user query, and (3) the desired output parameters. In shorthand, the interface can be written as \(R = (A_{top3}, P_{in}, P_{out})\). Given \(R\), TWNM performs graph search from each recalled target API to construct candidate toolchains, using the scoring function in Appendix B.2 to guide expansion and pruning. The selected subgraphs are returned to the LLM not as raw graphs but as serialized dependency trees, as described in Sec. 3.1.1. A simplified example is:
> ```text
> TargetAPI
> ├── parameter_a
> ├── parameter_b
> └── parameter_c
>     └── SupportingAPI
>         └── parameter_d
> ```
> We will make this interface and return format more explicit in the revised main text.
>
> **Q2: Evaluation of pruning/reactivation mechanisms**
>
> A2: We sampled 50 ToolBench queries and evaluated two consecutive phases on the same query set. In Phase 1, 10% of APIs were randomly selected to fail upon invocation; in Phase 2, these APIs were assumed to recover. After each query, we updated API failure and usage statistics, pruned APIs whose pruning scores exceeded 0.7 (3% of APIs in total), and randomly selected 10% of the pruned APIs for reactivation. We compared two settings: (1) no pruning/reactivation and (2) full NaviAgent. As shown in the table below, dynamic API maintenance improves both task success and efficiency under changing API availability.
>
> |Method|TSR(%)|Steps|
> |---|---|---|
> |w/o mechanisms|44|5.12|
> |NaviAgent|48|4.72|
>
> **Q3: What does “real time API execution is currently unavailable” mean, and what does it imply for evaluation validity?**
>
> A3: This does not affect the validity of our benchmark evaluation. It reflects an external platform limitation rather than a limitation of NaviAgent itself. Specifically, the current public ToolBench evaluation setup does not provide real-time API execution, and we follow the same benchmark setup and evaluation standard as prior work to ensure a fair and direct comparison. To further validate practical performance, we additionally evaluate NaviAgent on 50 real APIs from RapidAPI and report the results in Section 4.3.
>
> **Q4: What role does SFT play in NaviAgent?**
>
> A4: SFT is included to demonstrate the broad applicability of NaviAgent across model scales. This is particularly useful in resource constrained settings, where larger models may not be affordable. In such cases, the same framework can be deployed with a smaller backbone and further strengthened through supervised fine tuning. As discussed in Section 4.2 under Adaptability through Fine tuning and shown in Table 3, SFT substantially improves Qwen2.5 14B, bringing it to performance comparable to the larger 32B model.

---

> > ### Author Rebuttal · Reviewer_pvXC · 2026-04-04
> >
> > I appreciated the authors clarifications and additional evidence to address my concerns. I think these clarifications will improve the work, and I would stick with my original score of 5 - Accept.

---

> > > ### Author Response · Authors · 2026-04-04
> > >
> > > We are grateful for your positive feedback. We will revise the manuscript accordingly. Thank you again for recognizing our work.

---

### Official Review · Reviewer_acit · 2026-03-09

**Soundness:** 3
**Presentation:** 3
**Significance:** 3
**Originality:** 3
**Overall Recommendation:** 5
**Confidence:** 3

**Summary:**

This paper proposes NaviAgent, a graph-driven LLM agent framework for large-scale tool orchestration, which separates high-level planning from low-level tool execution and uses a dynamic tool graph to model API relations, retrieve toolchains, and adapt to API changes. Experiments on API-Bank, ToolBench, and a real-world API environment show that the method improves task success and robustness over strong baselines, especially in complex multi-tool settings.

**Compliance With Llm Reviewing Policy:**

Affirmed.

**Key Questions For Authors:**

1. How is the graph constructed in the initial cold-start stage? In particular, how many pre-executed historical trajectories or interaction records are used to initialize the graph?
2. How is a newly introduced tool incorporated into the graph? Does adding a new tool require retraining the model, or can the graph be updated incrementally? Also, has the paper considered how the degree of discrepancy between the new tool and existing in-graph tools affects this process?
3. Has the paper considered a training-free alternative, where an agent autonomously decides how to organize and use tools without relying on a separately trained model?

**Limitations:**

yes

**Strengths And Weaknesses:**

Strengths:
1. The method is well structured: it cleanly separates high-level decision-making from low-level tool execution, and the TWNM explicitly models structural and behavioral dependencies among tools.
2. The evaluation is relatively broad, covering not only API-Bank and ToolBench but also a real-world setting with 50 live APIs, which makes the empirical validation more convincing.
Weaknesses:
1. The overall framework is fairly complex, involving graph construction, graph modeling, dynamic evolution, and path recombination, while the discussion of engineering cost and maintenance overhead is relatively limited.
2. Figure 6 in the ablation study does not clearly demonstrate the performance impact of these components.
3. Given the complexity of the overall framework, the current ablation study is not sufficient to clearly disentangle the contribution of each component, including graph construction, graph modeling, dynamic evolution, and path recombination.

---

> ### Author Rebuttal · Authors · 2026-03-31
>
> **Q1: What engineering cost and maintenance overhead does the full framework introduce?**
>
> A1: The overhead is mainly offline or event-triggered, not per-query. TWNM is updated periodically rather than retrained after each change. At our current scale of 5.5K tools, 7.9K nodes, and 24K edges (Appendix G, Table 8), full offline retraining takes 94 s on a single NVIDIA P40 GPU and is hot-swapped with negligible serving latency. Incremental updates are lightweight: hot-node patches (updates of tool usage statistics) take about 55 ms, new-tool insertion (including semantic matching and attribute initialization) under 100 ms, and memory usage is about 1.5 GB on a 4-core CPU server with 8 GB RAM. Path recombination is only triggered after API failures. Overall, the maintenance overhead is modest in practice.
>
> **Q2: Current ablation study is not sufficient to clearly disentangle the contribution of each component.**
>
> A2: The paper already provides ablation studies for the graph-related modules in Section 4.4 (Table 3). In particular, the static graph improves over the no-graph baseline by about 6.5 TSR points on average, indicating the effectiveness of graph modeling. Building on this, the dynamic graph brings further gains, especially on hard tasks, and heuristic search further improves over the Alpha-Beta variant. To make the role of the bilevel design more explicit, we additionally conduct an ablation on the four-action decision space (see the table below). On a stratified sample of 100 queries using DeepSeek-V3, the full NaviAgent achieves the best TSR (53%), compared with 48% without clarification and 51% when retrieval and execution are merged. For path recombination, which is mainly triggered in failure-recovery cases (e.g., failed API calls), we provide additional results in our response to Reviewer pvXC Q2.
>
> |Method|TSR(%)|Steps|
> |---|---|---|
> |w/o Clarification|48|4.72|
> |w/ Merged Retrieval-Execution|51|4.46|
> |NaviAgent|53|4.67|
>
> **Q3: How is the graph constructed at cold start, and how many historical trajectories are needed for initialization?**
>
> A3: TWNM does not require historical trajectories to construct the cold-start graph. TWNM initializes the graph directly from tool schemas by linking APIs through matched input/output parameters, with inconsistent API/parameter names (e.g., latitude, lat, and geo_lat) normalized using semantic similarity over their descriptions. In our implementation, historical trajectories are only used to initialize the dynamic component: we bootstrap behavioral dependencies and edge weights from 70% of the deduplicated trajectories and hold out the rest for evaluation. Importantly, Table 2 (Sec. 4.3) shows that even without historical usage data, the static graph variant (Static+A) still outperforms the no-graph baseline by about 6.5 points on average, indicating that the structural graph alone is already effective at cold start.
>
> **Q4: How are newly introduced tools incrementally integrated into the graph without retraining, especially when they differ from existing tools?**
>
> A4: As described in Sec. 3.2.4, TWNM incorporates a newly introduced tool through the Incremental Node Integration mechanism. The new tool is matched to existing ones using semantic similarity over tool and parameter descriptions. Its dynamic attributes (e.g., successful call counts) are initialized to zero, and its initial node embedding is set as a weighted average of its neighbors. Adding a new tool does not require immediate retraining; graph refresh is periodic (weekly in our current deployment). In our design, the degree of discrepancy directly determines how informative the insertion step can be: tools that are closer to existing ones can be attached with more reliable neighbors and priors, whereas more discrepant tools rely more on schema-level connections and are refined later through subsequent interaction feedback.
>
> **Q5: Has the paper considered a training-free way to organize and use tools, without relying on a separately trained model?**
>
> A5: We agree that a training-free alternative is an interesting direction. In this work, we focus on multi-tool settings where tasks require nontrivial tool composition. In such settings, relying solely on the LLM to infer tool relations at inference time may be insufficient. TWNM is introduced to maintain these relations explicitly, allowing the agent to focus on high-level decision making.

---

> > ### Author Rebuttal · Reviewer_acit · 2026-04-02
> >
> > I keep my original positive score unchanged.

---

> > > ### Author Response · Authors · 2026-04-02
> > >
> > > We sincerely appreciate your time and recognition of our work.

---

### Official Review · Reviewer_kbUY · 2026-03-12

**Soundness:** 3
**Presentation:** 3
**Significance:** 3
**Originality:** 3
**Overall Recommendation:** 4
**Confidence:** 3

**Summary:**

This paper proposes NaviAgent, a bilevel agent architecture that separates high-level decision planning from low-level tool execution through a graph-based Tool World Navigation Model. The framework models structural and behavioral dependencies among APIs using a heterogeneous graph and supports scalable multi-tool orchestration through graph navigation and dynamic path recombination. Experiments on ToolBench, API-Bank, and real-world APIs show notable improvements in task success rate compared to several agent baselines.

**Compliance With Llm Reviewing Policy:**

Affirmed.

**Final Justification:**

This paper presents a technically solid and reasonably original framework for scalable multi-tool orchestration, with strengths in the graph-based tool modeling, bilevel design, and broad empirical evaluation. The rebuttal helped clarify several points, especially the retrieval–search interaction, the reliability of TSR evaluation, and the behavior under sparse logs or unseen tools. My remaining concerns are mainly about cleaner attribution of gains across components and the overhead of maintaining/updating TWNM at larger scale, which were only partially addressed. Overall, the rebuttal reinforced rather than changed my view, and I maintain my original recommendation.

**Key Questions For Authors:**

1. I wonder how sensitive NaviAgent is to the quality and coverage of the historical invocation logs used to construct the tool graph. Would the method still work well if such logs are sparse or noisy?
2. How the system behaves when completely unseen tools are introduced without any historical edges. Can the graph-based navigation still find reasonable toolchains?
3. Could the authors clarify the computational overhead of maintaining and updating the TWNM graph, especially when scaling to tens of thousands of APIs?
4. Have the authors considered integrating learning-based planning (e.g., reinforcement learning over tool graphs) instead of relying mainly on heuristic search strategies?

**Limitations:**

yes

**Strengths And Weaknesses:**

Soundness

Positive:
1. I feel the overall technical pipeline is quite reasonable. The bilevel decomposition between planning and execution makes sense conceptually, and it aligns well with the typical separation between reasoning and tool interaction in LLM agents.
2. I like that the authors try to formalize the tool ecosystem as a heterogeneous graph and use a graph transformer for link prediction. Modeling both structural (API-parameter) and behavioral (invocation logs) dependencies is a solid idea and seems technically grounded.
3. The paper also includes several ablations (e.g., static vs dynamic graph, Alpha-Beta vs heuristic search) and shows consistent improvements across models and datasets, which helps support the empirical claims.

Negative:
1. Some key implementation details are unclear. For example, how exactly the toolchain retrieval interacts with the graph search during inference is not fully specified, which may make reproduction difficult.
2. The evaluation relies heavily on LLM-based judging for TSR, which might introduce bias. I would like to see additional evaluation signals such as deterministic API result checks or human verification.

Presentation

Positive:
1. Overall I think the paper is fairly easy to read. The high-level idea of separating planning and execution is clearly explained in the introduction and illustrated well in the architecture figure.
2. The diagrams (e.g., the workflow figure and the graph evolution illustration) help readers understand how the system operates across planning, graph modeling, and execution loops.
3. The experimental section is relatively comprehensive, covering multiple models, datasets, and ablations.

Negative:
1. The narrative around the bilevel architecture could be sharper. At times it feels similar to existing planner-executor frameworks, but the paper does not clearly emphasize what is fundamentally different.

Significance

Positive:
1. The authors strive to focus on a significant problem: scaling LLM agents to large tool ecosystems with thousands of APIs. This is indeed an important challenge for real-world agent deployment.
2. I think the idea of explicitly modeling tool relationships with a graph and using navigation-based planning could inspire further research on structured tool reasoning.

Negative:
1. While the problem is important, the actual performance improvements are moderate in some cases and still leave relatively low absolute TSR numbers for complex tasks.
2. It is not entirely clear whether the gains mainly come from better tool retrieval, graph search heuristics, or the bilevel design itself. The contributions feel somewhat mixed together.

Originality

Positive:
1. The manuscript's important contribution concerns combining graph-based dependency modeling with a bilevel planning architecture for tool orchestration. This integration is interesting and not widely explored yet.
2. I appreciate the attempt to incorporate both structural API schemas and behavioral invocation data into a unified representation.
3. The dynamic graph evolution and path recombination mechanisms are thoughtful additions that address real-world API changes.

Negative:
1. Many components individually resemble prior work: planner–executor agents, tool graphs, link prediction, and heuristic search. The novelty mainly comes from combining them rather than introducing a fundamentally new paradigm.
2. The four-action decision space (direct response, clarification, retrieval, execution) is intuitive but not particularly novel compared to existing agent frameworks.

---

> ### Author Rebuttal · Authors · 2026-03-31
>
> **Q1: How does toolchain retrieval interact with graph search during inference?**
>
> A1: During inference, the agent first generates a tool description for the current task and uses it to retrieve the top-3 candidate target APIs from the full tool pool. These APIs are then used as roots for TWNM graph search, conditioned on the identified input and target output parameters. TWNM expands and prunes candidate paths using the scoring function in Appendix B.2, and the resulting subgraphs are serialized into tree-structured text for the agent context (Sec. 3.1.1). Additional details on the interface format and returned dependency trees are provided in our response to Reviewer pvXC Q1.
>
> **Q2: TSR relies on LLM judging. Any deterministic checks or human verification?**
>
> A2: For tasks with deterministic outcomes, we use exact-match evaluation. For example, when the answer is Yes/No, a number, or a specific string, it must exactly match the ground truth. For open-ended results, we use GPT-4.1 as the judge. To verify its reliability, we randomly sampled 3 groups of outputs, with 60 instances in each group (180 in total), and manually checked them by human evaluators. The agreement rates between GPT-4 and human judgments were 98.3%, 95.0%, and 96.7%, respectively, indicating that the overall evaluation protocol is reliable.
>
> **Q3: What is different between NaviAgent’s bilevel design and a standard planner-executor framework?**
>
> A3: In many planner-executor settings, the planner decomposes the task into solution steps and the executor carries them out. NaviAgent’s upper level does not plan how tools should be chained; it decides which interaction mode to invoke next, separating high-level reasoning from tool complexity. Its lower level is not an executor that merely carries out a pre-specified plan; it operates over TWNM to retrieve, search, and recombine tool paths online.
>
> **Q4: Why are the gains only moderate in some settings, and why does TSR remain low on hard tasks?**
>
> A4: The relatively low absolute TSR on complex tasks highlights that end-to-end tool-use remains an important and challenging problem. For tasks requiring long tool chains, any intermediate error can cause the final trajectory to fail. In this setting (see table 1), NaviAgent still yields consistent overall gains across all three backbones, improving TSR by about 10 absolute points on average (26.9 → 35.8, 32.8 → 45.4, and 40.7 → 55.2). The gain is smaller on the hard split for Qwen2.5-14B, likely because smaller backbones are more bottlenecked by long-horizon reasoning and recovery, but the improvement remains positive.
>
> **Q5: Unclear attribution of gains across the bilevel design, graph-based retrieval/planning, and search heuristics.**
>
> A5: Please see our response to Reviewer acit Q2.
>
> **Q6: What is the key novelty of NaviAgent?**
>
> A6: NaviAgent presents a new structural view of realistic tool use: instead of treating it as flat tool selection, it organizes tool use as decision making over interaction modes and navigation over tool dependencies. This creates a structured connection between task-level handling and toolchain construction, making retrieval, graph search, and recombination parts of the same toolchain-construction process rather than a loose combination of modules.
>
> **Q7: The four-action decision space is intuitive but not particularly novel compared to existing agent frameworks.**
>
> A7: Our four-action design differs from prior frameworks such as α-UMI and HuggingGPT by replacing a fixed pipeline with dynamic action selection. NaviAgent can choose to respond, clarify intent, retrieve tools, or execute tools at each step, and can re-retrieve during replanning when current tools are unsuitable.
>
> **Q8: Would the method still work well if such logs are sparse or noisy?**
>
> A8: NaviAgent is not solely dependent on historical invocation logs. While sparse or noisy logs may reduce the quality of the initial graph, it also leverages static structural information, especially dependencies induced by tool input/output parameters, and further updates the graph with newly generated calls during execution.
>
> **Q9: How does the system handle completely unseen tools without historical edges?**
>
> A9: As shown in Table 2 (Sec. 4.3), the static graph variant (Static+A) still outperforms the no-graph baseline by about 6.5 points on average, indicating that the structural graph alone is already effective at cold start.
>
> **Q10: What is the overhead of maintaining and updating the TWNM?**
>
> A10: Please see our response to Reviewer acit Q1.
>
> **Q11: Has learning-based planning been considered for NaviAgent?**
>
> A11: We see learning-based planning as a valuable extension of NaviAgent. In this work, we focus on heuristic search because it is training-free, interpretable, and easier to adapt to evolving tool graphs and new tools.

---

> > ### Author Rebuttal · Reviewer_kbUY · 2026-04-02
> >
> > Thank you for the detailed rebuttal. The clarification of retrieval–search interaction, the added exact-match/human verification for TSR, and the discussion of sparse logs and unseen tools are helpful, but the core attribution of gains across the bilevel design versus graph/search components and the overhead of maintaining/updating TWNM at larger scale remain insufficiently resolved. My main follow-up question is whether you can quantify graph maintenance/update cost separately from inference runtime and more cleanly isolate which component drives the reported gains. Based on the rebuttal, I would keep my original postive score unchanged.

---

> > > ### Author Response · Authors · 2026-04-03
> > >
> > > Thank you for the constructive feedback. Below we address your follow‑up questions in detail.
> > >
> > > **Q1: Clarification about the contribution of each component.**
> > >
> > > A1: We ran three additional ablations with DeepSeek‑V3 under the same setup as the first‑round rebuttal, including: (1) ReAct + Graph + Alpha (graph+search only), (2) Bilevel (bilevel only), (3) Bilevel + Graph + Unpruned (bilevel+
> > > graph without any search strategy, expanding all tool paths up to depth k=3). As shown in the table below, the graph module shows the largest improvement (+10pp, 35→45%), confirming that the dynamic graph structure capturing both tool relations and invocation behavior substantially enhances retrieval and reasoning. The bilevel design yields a comparable improvement (+8pp, 35→43%), serving as the main architectural contribution. Combining both raises TSR to 49–53%, with graph search progressing from unpruned to alpha‑beta to heuristic, providing an additional+4pp improvement. These results show that the three modules offer complementary rather than redundant gains.
> > >
> > > |Method|Bilevel Design|Tool Graph|Graph Search|TSR(%)|Steps|
> > > |---|---|---|---|---|---|
> > > |React|-|-|-|35|3.65|
> > > |React + Graph + Alpha|-|✓|Alpha-Beta|45|4.35|
> > > |Bilevel|✓|-|-|43|4.63|
> > > |Bilevel + Graph + Unpruned|✓|✓|Unpruned|49|5.77|
> > > |Bilevel + Graph + Alpha|✓|✓|Alpha-Beta|52|4.86|
> > > |Bilevel + Graph + Heuristic|✓|✓|Heuristic|53|4.67|
> > >
> > > **Q2: Clarification about graph maintenance/update cost.**
> > >
> > > A2: All maintenance measurements were obtained independently of inference runtime, and each operation runs asynchronously without affecting query latency. Specifically, graph maintenance involves three types of operations:
> > > (1) Node attributes are updated asynchronously in real time upon each successful or failed tool invocation. These updates are lightweight (55 ms per event) and modify only local node attributes without reloading the entire graph.
> > > (2) New tool insertions are also handled asynchronously in real time, taking under 100 ms per tool including semantic matching and attribute initialization; such events occur infrequently compared with the total number of queries.
> > > (3) Full graph retraining is periodically performed offline (94 s on a single NVIDIA P40) on a separate copy and asynchronously hot‑swapped into production, ensuring continuous availability.
> > > The average end‑to‑end inference runtime is 36 s per query, indicating that graph‑maintenance overhead is negligible in practice.
> > >
> > > We hope these results and clarifications address your remaining concerns. Thank you again for your time and thoughtful review.

---

### Official Review · Reviewer_14WR · 2026-03-12

**Soundness:** 2
**Presentation:** 3
**Significance:** 2
**Originality:** 2
**Overall Recommendation:** 3
**Confidence:** 4

**Summary:**

This paper proposes NaviAgent, a bilevel framework for LLM-based tool orchestration that decouples high-level task planning from low-level tool execution. The key contribution is the Tool World Navigation Model (TWNM), a heterogeneous graph transformer (HGT) that jointly models structural and behavioral dependencies among APIs via link prediction, enabling scalable toolchain retrieval and fault-tolerant path recombination. A four-action decision space (direct response, intent clarification, toolchain retrieval, tool execution) governs the planning level, while closed-loop feedback from real executions continuously updates the TWNM's edge weights. Evaluations on API-Bank and ToolBench, as well as 50 live RapidAPI endpoints, report consistent improvements in task success rate (TSR) over ReAct, ToolLLM, and α-UMI baselines.

**Compliance With Llm Reviewing Policy:**

Affirmed.

**Key Questions For Authors:**

Can you provide a direct ablation comparing TWNM against ToolNet's graph formulation on identical experimental settings?
  How does the live API evaluation (Table 2) compare to a simulation-based evaluation on the same 303 queries? This would quantify the simulator fidelity gap.
 Why was Tool-Planner (ICLR 2025) excluded from Table 1 despite being cited as related work and evaluated on ToolBench?
Have you tested NaviAgent's graph construction and inference at scales beyond 5,500 tools? What is the empirical scaling behavior?

**Limitations:**

Yes

**Strengths And Weaknesses:**

S1. Timely and well-motivated problem framing. The observation that existing tool agents lack a global relational view of inter-tool dependencies—leading to brittle sequential invocation—is a legitimate and practically important gap. The dichotomy between "structured but static" (e.g., ControlLLM) and "adaptive but unstructured" (e.g., ReAct-based methods) is a crisp framing that motivates the bilevel design.

S2. Genuine closed-loop adaptation. Unlike static graph approaches (e.g., ControlLLM, ToolNet), NaviAgent's edge attribute propagation mechanism (Eq. 14) blends long-term historical weights with a recency-weighted sliding window, giving the system the ability to gracefully degrade deprecated APIs and surface newly reliable ones. This is a practically significant property in production API ecosystems.

S3. Comprehensive experiments. The ablation study systematically isolates the contributions of (i) the base agent decision space, (ii) static vs. dynamic graph, and (iii) Alpha-Beta vs. heuristic search. The real-world evaluation on 50 live RapidAPI endpoints is a commendable addition that bridges the simulation–reality gap present in most prior tool-learning evaluations.

W1. Insufficient differentiation from ToolNet (Liu et al., 2024b). This is the most significant concern. ToolNet already organizes tools into a directed weighted graph, uses edge weights derived from historical invocation patterns for navigation, and demonstrates scalability to thousands of tools on ToolBench. NaviAgent's core graph construction (Section 3.2.1) appears to extend ToolNet primarily by (a) adding parameter nodes to create a bipartite API–parameter structure, and (b) replacing static edge weights with a learned HGT. The paper acknowledges ToolNet as related work but dismisses it in one sentence ("limited by sparse multi-hop interaction data") without quantitative comparison in the ablation. This reviewer finds the gap in novelty from ToolNet underexplored, and asks: how does NaviAgent's TWNM compare with ToolNet's graph construction and navigation when evaluated on an equal footing—same backbone, same dataset, same evaluation protocol?

W2. Simulated API environment undermines benchmark validity. The authors acknowledge that "real-time API execution is currently unavailable" and that all ToolBench and API-Bank evaluations are conducted in a simulated environment using an LLM-based API simulator. This is a critical limitation inadequately emphasized in the main text. The simulator's priority strategy (historical matching > structural cloning > contextual simulation) means the evaluation loop is partially circular: tools that appear frequently in training trajectories will be simulated more faithfully, potentially inflating TSR for methods that reuse high-frequency toolchains—precisely NaviAgent's design bias. The live RapidAPI experiments (303 queries) partially address this but do not cover the ToolBench/API-Bank settings used for the ablation study. The authors should provide a principled analysis of how simulator fidelity interacts with reported TSR gains, or move the primary evaluation entirely to live endpoints.

W3. The four-action decision space lacks meaningful novelty justification. The four actions (direct response, intent clarification, toolchain retrieval, tool execution) bear strong resemblance to the planning/invocation/summarization decomposition in α-UMI (Shen et al., 2024) and the plan/select/execute/respond pipeline in HuggingGPT (Shen et al., 2023). Beyond relabeling stages and introducing the graph-based retrieval action, the paper provides no ablation isolating the benefit of the four-action structure independently of the TWNM. Figure 6 shows that "Normal" (single-pass) cases dominate across all complexity levels, suggesting clarification and re-retrieval contribute marginally in practice. What is the TSR of a three-action variant (removing intent clarification or merging retrieval and execution) relative to the full NaviAgent?


W4. Baselines are dated or underrepresented for a 2025/2026 submission. The comparison set (ReAct, ToolLLM, α-UMI) is drawn almost entirely from 2023. Several directly relevant and more recent methods are omitted.

W5. The KL-projection theorem is a minor contribution framed as a major one. Theorem H.2 shows that a hard rule of the form "if action pp p then action qq q" corresponds to the KL-minimal projection of the base policy onto the mechanism-constrained manifold. This is a clean result but a fairly direct corollary of standard information projection theory (e.g., Csiszár & Matúš, 2003). The proof occupies two pages of appendix for a result that can be established in a few lines given standard convex analysis. More importantly, the mechanism in NaviAgent is not simply a fixed binary transition rule but a dynamic graph-conditioned constraint, so the connection between Theorem H.2 and the actual TWNM-constrained policy remains informal. The authors should either tighten this theoretical connection or appropriately reframe the KL analysis as a motivating analogy rather than a formal justification.

---

> ### Author Rebuttal · Authors · 2026-03-31
>
> **Q1: Comparison with ToolNet and Tool-Planner**
>
> A1: We added direct comparisons with ToolNet (2024) and Tool-Planner (2025) using DeepSeek-V3 on 100 ToolBench queries, sampled from the easy, medium, and hard subsets via stratified sampling. As shown in the table below, NaviAgent achieves both higher TSR and substantially fewer steps than both methods. The reduction in steps is consistent with the fact that ToolNet and Tool-Planner perform step-by-step tool selection, whereas NaviAgent uses its graph structure to retrieve a global toolchain. The higher TSR is further supported by the comparison between NaviAgent (Tool-Only Static+A) and NaviAgent (Static+A), where TSR improves from 46 to 50, suggesting that the main gain comes from explicitly modeling inter-tool dependencies through input/output parameters. Compared with ToolNet’s trajectory-based relations and Tool-Planner’s toolkit organization, this richer dependency modeling better explains NaviAgent’s advantage.
>
> |Method|TSR(%)|Steps|
> |---|---|---|
> |toolplanner|43|8.37|
> |toolnet|44|5.95|
> |NaviAgent(Tool-Only Static+A)|46|4.55|
> |NaviAgent(Static+A)|50|4.69|
> |NaviAgent|53|4.67|
>
>
> **Q2: How is the validity of the simulated environment assessed?**
>
> A2: We agree that simulator fidelity is an important limitation and will emphasize it more clearly in the paper. API-Bank is a mock-tool benchmark by design. For ToolBench, although tools were originally intended for real API invocation, execution was no longer available when we conducted our experiments, making fully live reproduction infeasible. Some queries are also hard to evaluate with real-time APIs because their outputs are time-varying or state-dependent, reducing reproducibility. To mitigate concerns about historical trajectory leakage, we construct the tool graph and compute graph frequency statistics from only 70% of the deduplicated historical trajectories; evaluation queries are constructed from the remaining 30%. Thus, the test set is not a direct replay of the graph-construction data. Although this gap cannot be directly quantified because the benchmark and live API studies are not one-to-one matched in tools and queries, the live API results in the paper still suggest that the gains are not purely artifacts of the simulated setting.
>
> **Q3: Effectiveness of the Four-Action Decision Space**
>
> A3: NaviAgent differs from prior frameworks such as α-UMI and HuggingGPT by allowing retrieval and tool use to be decided dynamically rather than through a fixed pipeline. At each step, it can choose to respond, clarify intent, retrieve tools, or execute tools. This is especially useful in large tool ecosystems with graph-structured dependencies, where the agent may need to retrieve again during replanning if current tools are unsuitable. We have added ablation experiments on the four-action design using DeepSeek-V3 on 100 queries in total, drawn from the easy, medium, and hard subsets via stratified sampling. The full NaviAgent achieves the best TSR (53%), compared with 48% when intent clarification is removed and 51% when retrieval and execution are merged. This shows that explicitly modeling these decision modes improves performance. Clarification and re-retrieval mainly address cases with missing user inputs or unusable initial retrieval results; although less frequent, they are often necessary for successfully completing complex tasks.
>
> |Method|TSR(%)|Steps|
> |---|---|---|
> |w/o Clarification|48|4.72|
> |w/ Merged Retrieval-Execution|51|4.46|
> |NaviAgent|53|4.67|
>
> **Q4: Is the KL-projection theorem sufficiently connected to the actual TWNM-constrained policy?**
>
> A4: We agree that this part could be stated more carefully. Our intent was not to claim a full characterization of the TWNM-constrained NaviAgent policy, but to provide an intuitive variational view of inference-time mechanism injection as a minimal KL correction under local feasibility constraints. In the revision, we will tone this down and generalize the result from the singleton hard rule (“if $p,$ then $q$”) to a context-dependent feasible action set $\mathcal{A}_{\mathrm{feas}}(h).$
>
> **Q5: What is the empirical scaling behavior of NaviAgent?**
>
> A5: ToolBench already provides a large-scale setting with 5,501 APIs, which is also the full scale used in our main experiments. To further assess scalability within this benchmark, we tested NaviAgent at 1k, 3k, and 5.5k APIs. As shown in the table below, the graph grows from 2.4k to 7.8k nodes and from 5.3k to 24.2k edges, while TSR decreases only moderately (59 → 54) and Steps remain stable. These results suggest stable inference efficiency with only a moderate performance drop as the tool space expands.
>
> |APIs|Nodes|Edges|TSR(%)|Steps|
> |---|---|---|---|---|
> |1k|2.4k|5.3k|59|4.76|
> |3k|5.4k|13.7k|57|4.48|
> |5.5k|7.8k|24.2k|54|4.62|

---

> > ### Author Rebuttal · Reviewer_14WR · 2026-04-05
> >
> > Thank you for your response and clarification. As mentioned by the authors, Q2 and Q4 are still there and thus I will keep my score.

---

> > > ### Author Response · Authors · 2026-04-06
> > >
> > > Thank you for the additional feedback. We address Q2 and Q4 directly below.
> > >
> > > **Q2: Clarification about the validity of simulated API environment**
> > > A2: We completely agree that real-world applicability is the ultimate goal. Large-scale live evaluation typically involves trade-offs among reproducibility, cost, safety, and practicality [1]. The live and simulated API sets are also not directly aligned, and the live environment itself is unstable. StableToolBench [2] reports that “the stability of API status for a significant portion of online tools (55.6%) is inconsistent,” which “undermines the reliability and reproducibility of model performance assessments.” Likewise, WebArena [3] highlights “unpredictable modifications” in live content, making controlled large-scale evaluation difficult. Therefore, a rigorous quantitative fidelity analysis between simulated and live settings would be difficult to interpret, because the observed gap would conflate simulation fidelity with uncontrolled variation in the live environment.
> > >
> > > At the same time, our comparison remains fair across methods because the baselines, including $\alpha$-UMI and ToolPlanner, are evaluated under the same ToolBench simulated environment. Such benchmarks are intended to assess core tool-use abilities, including intent understanding, tool selection, parameter extraction, and multi-step planning, rather than transient factors such as network connectivity or changing external service status. To further assess real-world transfer and robustness, the paper already includes live RapidAPI experiments, and we additionally provide simulated API unavailability tests (see our response to Reviewer pvXC, Q2). These results help verify that the method remains effective beyond the standard simulated benchmark.
> > >
> > > [1] Yang, Rui et al. EmbodiedBench: Comprehensive Benchmarking Multi-modal Large Language Models for Vision-Driven Embodied Agents. ICML 2025
> > >
> > > [2] Guo, Zhicheng et al. “StableToolBench: Towards Stable Large-Scale Benchmarking on Tool Learning of Large Language Models.” Annual Meeting of the Association for Computational Linguistics (2024).
> > >
> > > [3] Zhou, Shuyan et al. WebArena: A Realistic Web Environment for Building Autonomous Agents. ICLR 2024
> > >
> > > **Q4: Clarification about the KL-projection theorem**
> > > A4: We understand that the key concern is not the validity of the KL-projection statement itself, but whether it connects in a sufficiently direct and general way to the actual TWNM-conditioned policy in NaviAgent. Concretely, the original version used a singleton hard rule, forcing a fixed action $q$ in certain contexts. This was too special-case and weakened its connection to NaviAgent’s graph-conditioned mechanism. In the revision, we generalize it to a context-dependent admissibility formulation, where each context $h$ induces an admissible action set.
> > > $$\mathcal A_{\mathrm{feas}}(h)\subset \mathcal A,$$
> > > and we define the feasible policy class by
> > > $$\Pi_{feas}:=\left\\{\pi:\operatorname{supp}(\pi(\cdot| h))\subseteq\mathcal A_{\mathrm{feas}}(h),\ \forall h\right\\}.$$
> > > Under a context distribution $\mu$, we then consider the KL-minimal feasible correction of a base policy $\pi_0$:
> > > $$\pi_{\mathrm{inj}}
> > > \in
> > > \arg\min_{\pi\in\Pi_{\mathrm{feas}}}
> > > \mathbb E_{h\sim\mu}
> > > \left[
> > > D_{\mathrm{KL}}\bigl(\pi(\cdot\mid h),|,\pi_0(\cdot\mid h)\bigr)
> > > \right].$$
> > > The revised manuscript now gives the following more general theorem: if for every context $h$ with positive probability, the feasible set $\mathcal A_{\mathrm{feas}}(h)$ is nonempty and the base policy assigns positive mass to that set, i.e.,
> > > $$\sum_{a\in \mathcal A_{\mathrm{feas}}(h)} \pi_0(a\mid h)>0,$$
> > > then the above optimization problem has a unique solution, given explicitly by
> > > $$\pi_{\mathrm{inj}}(a\mid h)\frac{
> > > \pi_0(a\mid h)\mathbf 1_{a\in \mathcal A_{\mathrm{feas}}(h)}
> > > }{
> > > \sum_{a'\in \mathcal A_{\mathrm{feas}}(h)} \pi_0(a'\mid h)
> > > }.$$
> > > That is, mechanism injection precisely corresponds to restricting the base policy to the admissible action set and renormalizing; equivalently, it is the information projection of the base policy onto the admissible-policy class.
> > >
> > > This revised formulation is directly aligned with the actual decision structure of NaviAgent. In our full framework, the decision context at time $t$ is
> > >
> > > $$h_t=(H_t,O_t,{G}_{t-1}'^*),$$
> > >
> > > where the history, observation, and graph state jointly determine the admissible actions. Thus, graph pruning, substitution, rerouting, and subgraph switching can all be viewed as changing $\mathcal A_{\mathrm{feas}}^*(h_t)$. This makes the revised theorem a more direct and general match to NaviAgent/TWNM than the original toy rule.
> > >
> > > From the outset, the paper did not position this result as a main contribution, but included it as an interpretive lens for mechanism injection. In the revision, we have made this positioning more explicit and more measured.
> > >
> > > We hope the points above help clarify our response. Thank you for your time and consideration.

---

### Decision · Program_Chairs · 2026-04-30

**Decision:**

Accept (regular)

**Comment:**

The paper proposes NaviAgent, a tool orchestration architecture that utilizes graph-based modeling for the planner and the executor. Reviewers acknowledge the novelties of the paper, such as the modeling of tool use dependency as a dependency graph. The authors further addresses some concerns from the reviewers. Overall the paper meets the level of ICML. The authors should address the reviewers concerns in a thoroughly revised version. I would like to single out one particular concern from reviewer 14WR on the validity of simulated API environment. I share this concern, and hope that the authors could try their best to compensate their main experiments on the simulated environment with some real-world environments in their final version.